# The attitudes, perceptions and experiences of medical school applicants following the closure of schools and cancellation of public examinations in 2020 due to the COVID-19 pandemic: a cross-sectional questionnaire study of UK medical applicants

Katherine Woolf  , David Harrison, Chris McManus

► Prepublication history and additional materials for this paper is available online. To view these files, please visit the journal online (http://dx.doi.org/10.1136/bmjopen-2020-044753).

Research Department for Medical Education, University College London Medical School, London, UK

**Correspondence to**
Dr Katherine Woolf;
k.woolf@ucl.ac.uk

## ABSTRACT

**Objective** Describe the experiences and views of medical applicants from diverse social backgrounds following the closure of schools and universities and the cancellation of public examinations in the UK due to COVID-19.

**Design** Cross-sectional questionnaire study, part of the longitudinal UK Medical Applicant Cohort Study (UKMACS).

**Setting** UK medical school admissions in 2020.

**Participants** 2887 participants completed an online questionnaire from 8 April to 22 April 2020. Eligible participants had registered to take the University Clinical Admissions Test in 2019 and agreed to be invited to take part, or had completed a previous UKMACS questionnaire, had been seriously considering applying to medicine in the UK for entry in 2020, and were UK residents.

**Main outcome measures** Views on calculated grades, views on medical school admissions and teaching in 2020 and 2021, reported experiences of education during the national lockdown.

**Results** Respondents were concerned about the calculated grades that replaced A-level examinations: female and Black Asian and Minority Ethnic applicants felt teachers would find it difficult to grade and rank students accurately, and applicants from non-selective state schools and living in deprived areas had concerns about the standardisation process. Calculated grades were generally not considered fair enough to use in selection, but were considered fair enough to use in combination with other measures including interview and aptitude test scores. Respondents from non-selective state (public) schools reported less access to educational resources compared with private/selective school pupils, less online teaching in real time and less time studying during lockdown.

**Conclusions** The COVID-19 pandemic has and will have significant and long-term impacts on the selection, education and performance of our medical workforce. It is important that the views and experiences of applicants from diverse backgrounds are considered in decisions affecting their future and the future of the profession.

### Strengths and limitations of this study

► This is the first systematic exploration of medical applicant views on and experiences of the most significant changes to UK education in living memory due to the SARS-CoV-2/COVID-19 pandemic.

► It is also the first study we are aware of that looked at university applicant views on calculated grades and the perceived impact on university admissions this year and in 2021.

► The large sample size gathered from around the UK, and the richness of the data provide insight into differences in the experiences and views of different sociodemographic groups, after controlling statistically for educational attainment.

► It is uncertain how representative our sample is of all medical applicants; medical applicants are not representative of all university applicants in either academic or sociodemographic terms and generalisations from our findings to all applicants should only be done with extreme caution.

► At the time of writing it was not possible to include data on participant examination scores or applications and acceptances to medical school; however this follow-up is planned.

## INTRODUCTION

The UK Medical Applicant Cohort Study (UKMACS) is a study of UK medical school admissions. It is primarily a longitudinal questionnaire study of UK residents who in the summer and autumn of 2019 were seriously considering applying to study medicine in the UK for entry in 2020. UKMACS questionnaire data are subsequently linked to administrative data on all UK medical applicants held within the UK Medical Education Database (www.ukmed.ac.uk). Wave 1 data were collected

between May and September 2019 and asked how applicants from different backgrounds were choosing which medical schools to apply to. Wave 2 data were collected from November 2019 to January 2020 and asked which medical schools and universities participants had applied to and how they had made their choices.

In March 2020 it was announced that UK schools would close and A-level (and equivalent public examinations) would be cancelled due to the COVID-19 outbreak in the UK. This was one of the most major disruptions ever to affect education and university admissions in the UK and was very significant for the UKMACS cohort, who are mostly in their final year of schooling and were due to sit examinations in the summer of 2020.

We therefore administered an additional unplanned UKMACS questionnaire to understand what medical applicants were experiencing in terms of education, their views on how grades would be awarded following examination cancellations and their views on how medical schools might respond with regard to admissions policies. We particularly sought to understand how applicants from diverse social backgrounds might differ, with the aim of facilitating the inclusion of applicant perspectives and experiences in discussions about changes to medical school admissions and medical education.[1]

## Calculated grades

The absence of A-levels and other equivalent public examinations in March 2020 meant that alternative methods of assessment for candidates had to be found, not least as A-levels are 'the single most important bit of information (used in selection)' by universities.[2] On 3 April *Ofqual* (Office of Qualifications and Examinations Regulation) in England announced that exams under its purview in England would be replaced by calculated grades based on teachers' estimation of the grades that their students would have attained and the ranking of each student within grades (eg, if a teacher has 30 Chemistry A-level students, they would estimate the grade each student would get. Then the teachers rank students within grades, so eg, if they have five students estimated to get an A grade, they rank those five students), which would then be standardised centrally.[3] The Scottish Qualification Authority and other national bodies also announced similar processes for their examinations.

Performance in A-level examinations has long-term impacts,[4 5] which makes changes to how grades are awarded potentially very significant. The use of calculated grades raises many questions, some of which were summarised in a letter to *The Guardian* by Yasmin Hussein, a GCSE (General Certificate of Secondary Education) student who said that,

> … the … exam hall (is) a level playing field for all abilities, races and genders to get the grades they truly worked hard for and in true anonymity (as the examiners marking don't know you). […… Now we] are being given grades based on mere predictions.

Yasmin Hussein, letter to The Guardian, 29 March 2020.[6]

Among teachers, survey data suggest that there are doubts about the accuracy and fairness of calculated grades, with 39% saying that all students would get a fair deal, 24% saying they would not, and 37% not knowing or not answering. There were also doubts about fairness for students from Black Asian and Minority Ethnic (BAME) backgrounds, about those working hard in the last weeks before an exam being penalised, about teacher 'favouritism', although there were teachers who commented that the process is as fair as possible under the circumstances.[7]

University applicants also have concerns. In a survey carried out by the Higher Education Policy Institute before the details of calculated grades were announced, but after it was known that grades would in some way be predicted, 27% thought that their predicted grades were worse than they were likely actually to have attained, compared with 13% thinking their predicted grades were better than they would actually attain.[8]

Another survey of 511 university applicants (including 452 A-level students) conducted for the Sutton Trust found that just under half believed the new A-level grading system would result in their receiving poorer grades but working class respondents were more worried about large negative consequences compared with middle class students. Nearly three quarters believed the new system was less fair than examination grades and this was more of a concern for applicants from *higher* socioeconomic backgrounds. Nearly half of applicants felt the COVID-19 crisis would impede their chances of getting into their first choice university, a more common concern among working-class respondents.[9]

The impact on medical school admissions of examination cancellations and their replacement with calculated grades is, at the time of writing, still not completely clear. *Ofqual* stated that,

> The grades awarded to students will have equal status to the grades awarded in other years and should be treated in this way by universities, colleges and employers. On the results slips and certificates, grades will be reported in the same way as in previous years,[3] p.6.

The decisions of *Ofqual* in this case are in effect governmental decrees, supported by Ministerial statement, and universities and other bodies will therefore abide by them, as was affirmed by the Medical Schools Council on 5 May 2020.[10] That does not mean however that other factors were not needed to be taken into account in some cases, as for instance, when applicants did not attain the grades needed for their conditional offers, or for applicants in clearing. Furthermore in guidance updated on 1 May 2020 the Government stated that 'if a student does not feel their grade reflects their performance, they will have the opportunity to take an exam in the autumn'[11] with *Ofqual* expanding on 15 May 2020 that 'students will

be able to use the higher of the two grades for future progression'.[3] This raises questions for university admissions, as Medical Schools Council acknowledged in their statement of 5 May 2020:

> There are a number of issues that the education sector as a whole is yet to resolve. These include how appeals against calculated grades will work across the UK and when students will be able to sit exams if they are unhappy with their calculated grade. The impact of these issues on medical admissions is unclear but medical schools are actively engaging in these discussions and are working hard to develop solutions that are fair to applicants.[10]

### Education during the pandemic

As well as examinations being cancelled, UK schools closed on 20 March 2020 to all except the children of key workers and vulnerable children with secondary schools mostly closed until September 2020. Similarly in mid-March 2020 many universities suspended face-to-face teaching for the academic year 2019/2020.

The impact of school closures on student learning and outcomes will be significant[12–14] and it may be particularly problematic for those from poorer backgrounds and/or at state-funded schools. The Institute of Fiscal Studies (IFS) analysed survey data from a weighted sample of over 4000 parents with children aged between 4 years and 15 years in May 2020.[15] Among secondary school children, those from the richest quintile were spending on average slightly over an hour more per day on learning compared with those in the poorest quintile, amounting to several weeks more learning over the course of the time schools are closed. In particular children in the richest families were spending significantly more on educational activities provided by schools and from private tutors. Even among state school pupils, children from the richest families reported greater access to face-to-face online teaching, which the authors argue is likely to be of higher educational value than other resources that require more parent input, particularly since the poorest parents of secondary school children were less likely to find it easy to support their child's home learning.

The results of the IFS report chime with data from *Teacher Tapp*, an ongoing weighted survey of several thousand teachers in England.[16] At the start of the lockdown (23 March 2020) private secondary schools were much more likely than state secondary schools to be using online videoconferencing (27% vs 2%) and online chat (18% vs 3%). The above-mentioned Sutton Trust report[9] also found socioeconomic differences in access to 'internet access, devices for learning or a suitable place to study' and differences in the amount of A-level teaching being conducted by teachers at private and state schools.

Among those secondary school pupils who had applied to university, the Sutton Trust report authors argued that students from lower socioeconomic backgrounds are also likely to face additional disadvantages both with their university applications and when starting university:

> Given the uncertainty caused by these changes [to education resulting from COVID-19], university applicants are likely to need more support than ever to navigate the process (of applying to university). This will be even more important for young people from lower socio-economic backgrounds, who are less likely to be able to draw on the advice of family members with higher education experience themselves. But with schools closed for most pupils, it may be difficult for applicants to get the help they need. Similarly, there's also a danger that this year's applicants will miss out on A level content during the lockdown […]. For disadvantaged students about to go on to higher education, this could leave them with gaps in their knowledge base, putting them behind their peers before they have even begun at university.[9] p1

### The present study

This study aimed to explore and describe perceptions of calculated grades, of student selection more generally, and of educational experiences during school and university closures, in a large group of medical school applicants, who were typically high-attaining students. A range of background factors was assessed to determine how perceptions differed according to demographic and other measures. Data collection took place between 8 April and 22 April, which was about two and a half weeks after school closures.

## METHODS
### Study design

Cross-sectional questionnaire study, which formed part of the longitudinal UKMACS.

### Eligibility

To be invited to complete the questionnaire, participants had to have registered to take the University Clinical Admissions Test (UCAT) in 2019 and to have agreed to be invited to take part in UKMACS, or they needed to have completed one or more previous UKMACS questionnaires (Wave 1 of the UKMACS questionnaire was administered between May and October 2019; Wave 2 between November 2019 and January 2020.). They also need to have been seriously considering (Participants were thought to be seriously considering applying if they had registered to take UCAT. Wave 1 of the questionnaire also asked them to confirm they were seriously considering applying to study medicine.) applying to study medicine in the UK for entry in 2020, and be resident in the UK or Islands/Crown Dependencies.

Participants were not invited if they had previously requested their data be removed from the UKMACS database, had asked not to be contacted for further research, or had not consented to having their personal

information retained by the research team or linked with other information for research purposes.

## Questionnaire development

During the development of the questionnaire *Ofqual* announced that calculated grades would be awarded. We therefore assessed perceptions of how calculated grades would be awarded and used, and of other possible methods medical schools could use to select or reject offer holders. We also about the potential knock-on effects that calculated grades might have on the 2021 application cycle, and whether medical schools should open online or defer opening until teaching could be done face to face. We asked about use of educational resources and preparation for university/medical school, and about the time they were spending on various activities. We included self-reported measures of academic attainment and sociodemographic measures used in previous UKMACS questionnaires, as well as the 15-item Big Five personality measure used in the national longitudinal cohort study *Understanding Society.*[17] Personality traits are 'relatively enduring styles of thinking, feeling, and acting'.[18] It is generally agreed that there are five distinct personal traits or factors: Neuroticism, Extraversion, Openness to Experience, Agreeableness and Conscientiousness.

Most questions were designed specifically for this questionnaire since they asked about unprecedented events and validated items were not available. We constructed the questionnaire with JISC online surveys (https://www.onlinesurveys.ac.uk/) and piloted the questionnaire and information sheet with two current applicants to medical school. Amendments were made in response to feedback from the applicants and from Medical Schools Council. A copy of the questionnaire is included as online supplemental file 1.

## Questionnaire administration

Participants were sent an email invitation and link to the current questionnaire on the afternoon of 8 April 2020; 18,665 invitations were sent, with up to two email reminders and two text message reminders. The questionnaire closing date was 20 April 2020, with responses accepted up to 22 April 2020.

## Statistical analysis

Descriptive and univariate analyses were run on SPSS V.26. Imputation of missing data and multivariate analyses were run on R.

Factor analysis on the 87 attitudinal variables was carried out using the *psych* package in R[19] with *fa.parallel()* and *nfactors(),* being used to determine the number of factors.

## Freetext question answers

All answers to freetext questions were read by the research team, and illustrative quotes selected to aid understanding of quantitative results.

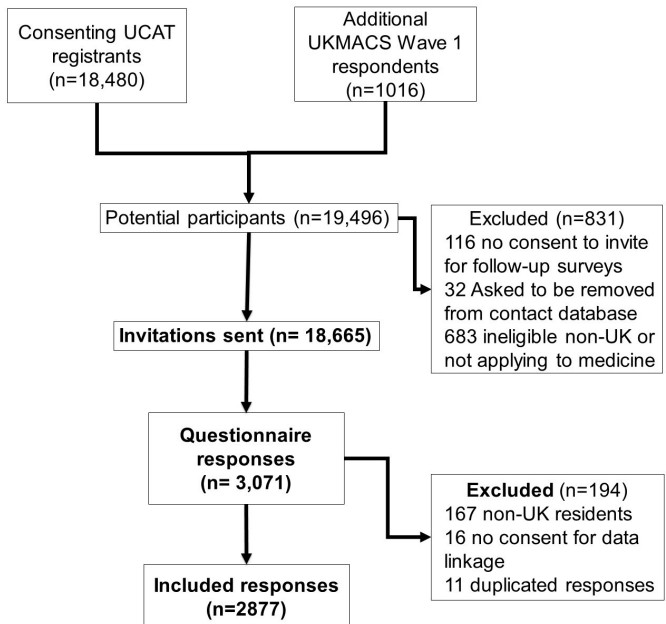

**Figure 1** Participant flow diagram. UCAT, University ClinicalAdmissions Test; UKMACS, UK Medical Applicant Cohort Study.

## Patient and public involvement

Patients and the public were consulted in the development of the questionnaire.

## RESULTS

### Participants

The questionnaire was completed by 3071 participants, of whom 2904 stated they were eligible to take part (ie, seriously considering applying to study medicine in the UK in 2020 and resident in the UK or Islands/Crown Dependencies). After removing 16 respondents who did not consent to have their data analysed and 11 duplicates, there were 2877 valid cases for analysis, which is 15% of those invited. This is subsequently referred to as the full sample (see figure 1).

The main analyses were performed on a restricted sample of 1562 respondents currently in Year 13, who had applied to medicine for entry in 2020, with at least three predicted A-levels and no achieved A-levels. Results are also reported in online supplemental file 2 for respondent groups excluded from the restricted sample, notably those living in Scotland and those not currently in Year 13 (see online supplemental file 2).

### Missing data

The analysis considered 120 measures in the restricted sample. The questionnaire asked about attitudes to 87 different topics concerning medical school entrance. Of 153 076 data points, 10 788 (7.2%) were missing. For the individual variables, the median percentage of missing data values was 0.48%, with 75 measures having fewer than 5% of missing values. The questionnaire also asked about demographic and educational items. For 12 demographic

measures, 462 of 18 744 measures were missing (2.5%), with a median of 1.0% per measure, and 11 measures having fewer than 5% missing values. For further details on missing data, please see online supplemental file 2.

## Demographics

Demographics for the full and restricted samples are reported in table 1, where details of categories within demographic variables can also be found.

## Education and achievement
### Predicted A-levels

A-level grades were scored as A*=12, A=10, B=8 and so on, and those reported as being between two grades as intermediate, for example, A*/A=11, A/B=9 and so on. Mean predicted A-level grades were calculated for the top three grades regardless of subject (*mean top three predicted A-levels*), and for all grades (*mean predicted A-levels*). *Mean top three predicted A-levels* was 10.89 and *mean predicted A-levels* was 10.71.

### Admissions test scores (UCAT, BMAT, GAMSAT)

Of the participants 1546 (99.1%) reported having taken the Universitys Clinical Aptitude Test (UCAT); 765 (49.0%) reported having taken the Biomedical Admissions Test (BMAT); and none reported having taken the Graduate Medical School Admissions Test (GAMSAT). Of the 1350 participants who reported a total UCAT score that was greater than 1799 and less than 3601, the mean score was 2660 (SD=235).

### General Certificate of Secondary Education (GCSE) grades

GCSE grades can range from 1 to 9. A variable *mean GCSE* was calculated by dividing the total GCSE points by the number of GCSEs taken, and the mean was 7.91 (SD=0.71).

### Relationships between educational measures

UCAT score correlated with *mean top three predicted A-levels* at 0.418 (p<0.001) and with *mean GCSE* at 0.487 (p<0.001). *Mean GCSE* and *mean top three predicted A-levels* correlated at 0.611 (p<0.001).

Participants at non-selective state schools had lower scores on all attainment measures (*mean GCSE*: difference=0.3 points, p<0.001; *mean top three predicted A-levels*: difference=0.23 points, p<0.001; UCAT score: difference=89 points, p<0.001).

## Medical school offers

Of the respondents 1292 (85%) had applied to four medical courses, 1289 (82.5%) had at least one offer, 177 (11.3%) had four offers and 204 (13%) were waiting to hear from at least one medical school at the time of completing the questionnaire.

Respondents who did not have a parent/carer with a university degree were less likely to have a medical offer (78.1% vs 85.0%; p=0.001), as were those without a parent/carer in the highest socioeconomic group (79%

**Table 1** Demographics for the full sample and the restricted sample (of those in Year 13, with at least three predicted A-levels, no achieved A-levels, who had applied to study medicine). Rounding to prevent identifying individuals

| | Full sample N (%) | Restricted sample N (%) |
|---|---|---|
| Female | 1968 (68) | 1097 (70) |
| Male | 749 (26) | 416 (27) |
| Other | 20 (<1%) | <10 (<1) |
| Missing | 140 (5) | Rounded to 40 (3) |
| White | 670 (23) | 516 (33) |
| Asian | 301 (11) | 228 (15) |
| Black | 79 (3) | 58 (4) |
| Mixed/other | 104 (4) | 87 (6) |
| Missing | 1723 (60) | 673 (43) |
| 1+ parents with degree | 1831 (64) | 1046 (67) |
| First in family | 895 (33) | 465 (30) |
| Missing | 151 (5) | 51 (3) |
| 1+ parents in the highest socioeconomic group | 1910 (66) | 1097 (70) |
| No parents in the highest socioeconomic group | 1742 (30) | 439 (28) |
| Missing | 116 (4) | 26 (2) |
| No parent doctors | 2408 (88) | 1334 (85) |
| 1+ parents who are doctors | 344 (13) | 192 (12) |
| Missing | 125 (4) | 36 (2) |
| Non-selective state school | 785 (27) | 590 (38) |
| Private or selective school | 783 (27) | 568 (36) |
| Missing | 1309 (46) | 404 (26) |
| IMD quintile 5 (most deprived—reverse scored) | 310 (11) | 169 (11) |
| IMD quintile 4 (reverse scored) | 361 (13) | 218 (14) |
| IMD quintile 3 (reverse scored) | 410 (14) | 236 (15) |
| IMD quintile 2 (reverse scored) | 461 (16) | 267 (17) |
| IMD quintile 1 (least deprived—reverse scored) | 704 (25) | 441 (28) |
| Missing | 631 (22) | 231 (15) |
| In Year 13/S6 | 2212 (77) | 1562 (100) |
| One year post-Year 13 | 179 (6) | 0 (0) |
| Have/studying for a degree | 340 (12) | 0 (0) |
| Mature without a degree/ other | 146 (5) | 0 (0) |
| Missing | 0 (0) | 0 (0) |
| England | 2003 (70) | 1281 (82) |
| Scotland | 170 (6) | <1 (<1) |
| Wales | 78 (3) | 50 (3) |

Continued

**Table 1** Continued

| | Full sample N (%) | Restricted sample N (%) |
|---|---|---|
| Northern Ireland/Forces/Islands | 66 (2) | Rounded to 40 (2) |
| Other/missing | 560 (20) | 192 (12) |
| Total | 2877 (100) | 1562 (100) |

IMD, Index of Multiple Deprivation.

vs 85%; p=0.002) Male participants were slightly less likely to have an offer (80% vs 85%; p=0.049).

### Applicant views on admissions
#### Perceptions of the fairness of methods medical schools could consider using in the selection of offer holders

Participants were asked to rate the fairness of 17 measures, including calculated grades, that medical schools could potentially use to decide to accept or reject an offer holder following exam cancellations. Rating categories were: 'Unfair: should not be used', 'Quite unfair: avoid if possible', 'Quite fair: could be used in combination with other measures' and 'Very fair: could be used alone', with a freetext question asking for additional comments and suggestions.

No measure was felt by a majority of participants to be fair enough to use on its own. The measure considered most fair was *Exam grades taken in September 2020 (if these take place)* (32.3% very fair), followed by *Predicted Grades declared on UCAS application* (26.2% very fair), *Calculated grades* (22.6% very fair), *GCSE grades* (20.4% very fair) and *Score at interview* (19.5% very fair) (see figure 2 for full item wording).

Several methods were felt by a majority to be fair enough in combination, particularly *Predicted grades* (80.6%), *GCSE grades* (73.8%) and *Score at interview* (73.4%); but only a fifth (20.3%) of participants felt *Attendance at widening participation activities* was quite fair or very fair (see figure 2).

Multiple regression results showed that after taking account of all other educational and sociodemographic variables, BAME participants were more likely to perceive *Exams taken in September 2020, UCAS personal statement* and *Personal background* as fair to use, and respondents from deprived areas were more likely to perceive *Personal background* and *Attendance at widening access programmes* as fair to use. *Calculated grades based on mock exams, coursework etc, and awarded in place of final examination grades* were perceived as less fair by those with lower predicted A-levels.

There were 154 freetext responses (10%), with participants elaborating on their responses or suggesting alternatives:

> A combination of the most objective information that every offer holder will have, that is, GCSEs, UCAT or BMAT, interview score, etc.

A standardised form of assessing all medical applicants would be the best way to allocate existing places. […] Since we do not have standardised A level grades, places should be offered using the UCAT as this is the fairest way of distributing places to the most able students.

> Using interview scores and UCAT scores in combination are independent measures, and are more fair than using calculated grades which have the potential to be biased.

> Anything including personal statement, BMAT or UCAT I would argue are unfair to use as judgement as there will definitely be a bias in terms of how certain students achieved their grade. I believe the fairest way to determine ones overall grade would be to use their GCSE data with a combination of evidence throughout the 2 years of A levels.

Other measures participants mentioned included: an additional university assessment (written, viva or project/portfolio-based) now or at the start of the academic year, an additional interview, selection at the end of Year 1/make first year a foundation year, additional reference from teachers/school, reference from work experience, school/college attendance record, distance from university, extenuating circumstances, self-reported use of time during quarantine/lockdown, number of offers received, prioritise those with higher degrees, prioritise those already working in the National Health Service (NHS), extracurricular achievement (eg, music, Duke of Edinburgh's Award), school's prior achievement. For example:

> NHS experience that is, patient facing health professional that is, years and grade, other non technical skills, education background that is, science, post graduate achievement that is, MSc particularly if in science or medical subject and grade achieved. Also emphasis on the candidates as a whole that is, well rounded personality (potential to communicate well) rather than typical A Grade student. Letter of recommendations from medical consultant whom candidates may have worked closely with.

> Another interview possibly over the phone to see what students have done with their time in quarantine (ie, volunteering in a care setting or hospital/working in a hospital/exploring other interests)

> Each university could form their own selection test similar to UCAT/BMAT with a brief guidance/specification on what will be on the test given out to offer holders so they have some time to revise for it, but this should be used in combination with other details (eg, if offer holder's calculated grade was only one grade below what was required for entry)

> I think a combination of previous results, any exams that do go ahead (at some point whether that is this summer or later), alongside medical applications, relevant work experience (as per personal statement

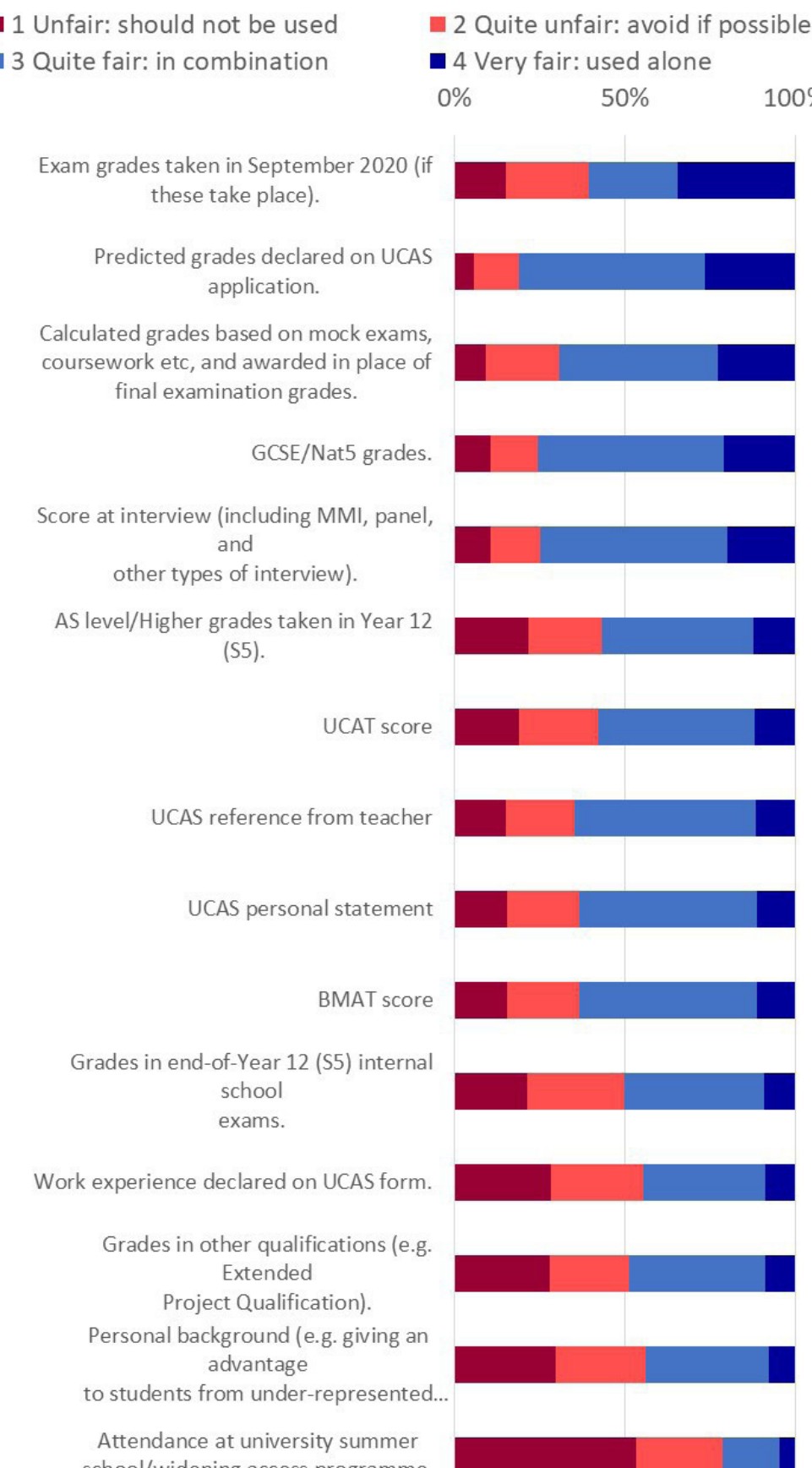

**Figure 2** Perceptions of the fairness of methods medical schools could use to decide whether or not to accept applicants who currently hold an offer now that exams have been cancelled. UCAT, University ClinicalAdmissions Test. MMI, Multiple Mini Interview. AS, Advanced Subsidiary.

and any other forms detailing this) and the applicant interview. Also potentially the medical schools could generate online admissions tests for students with conditional offers to generate a clearer view of a students capability and ability to comprehend and withstand the pressures of medical school. But any tests generated by the medical schools must be used alongside the other parts of the applications to ensure fairness.

Participants were asked whether they had heard anything from medical schools/universities they had applied to about how selection might be impacted by examination cancellations; among those holding conditional offers, a minority (n=538; 42%) said they had heard from at least one medical school/university they had applied to.

### Acceptability of options for dealing with a situation in which more students meet their offers than there are medical school places

Participants were asked to rate the acceptability ('completely unacceptable', 'slightly unacceptable', 'neutral', 'slightly acceptable', 'completely acceptable') of a number of options that medical schools could use if they had more students meeting offers than they had places, with a freetext question asking for additional comments and suggestions.

The most acceptable option was *Ask some applicants with offers to volunteer to defer a year*. The only other acceptable option was *Accept all applicants whose calculated grades meet the conditional offer, although it could mean fewer resources per student* (see figure 3).

Multiple regression analyses showed no significant differences by social or demographic group on these items.

There were 187 freetext responses (12%). Several respondents suggested that medical schools should receive more funding to manage larger cohorts and create more doctors, for example,

> Deferring of 1 year should not be taken into consideration as this would damage applications of next year. Ask the government to invest more money on the

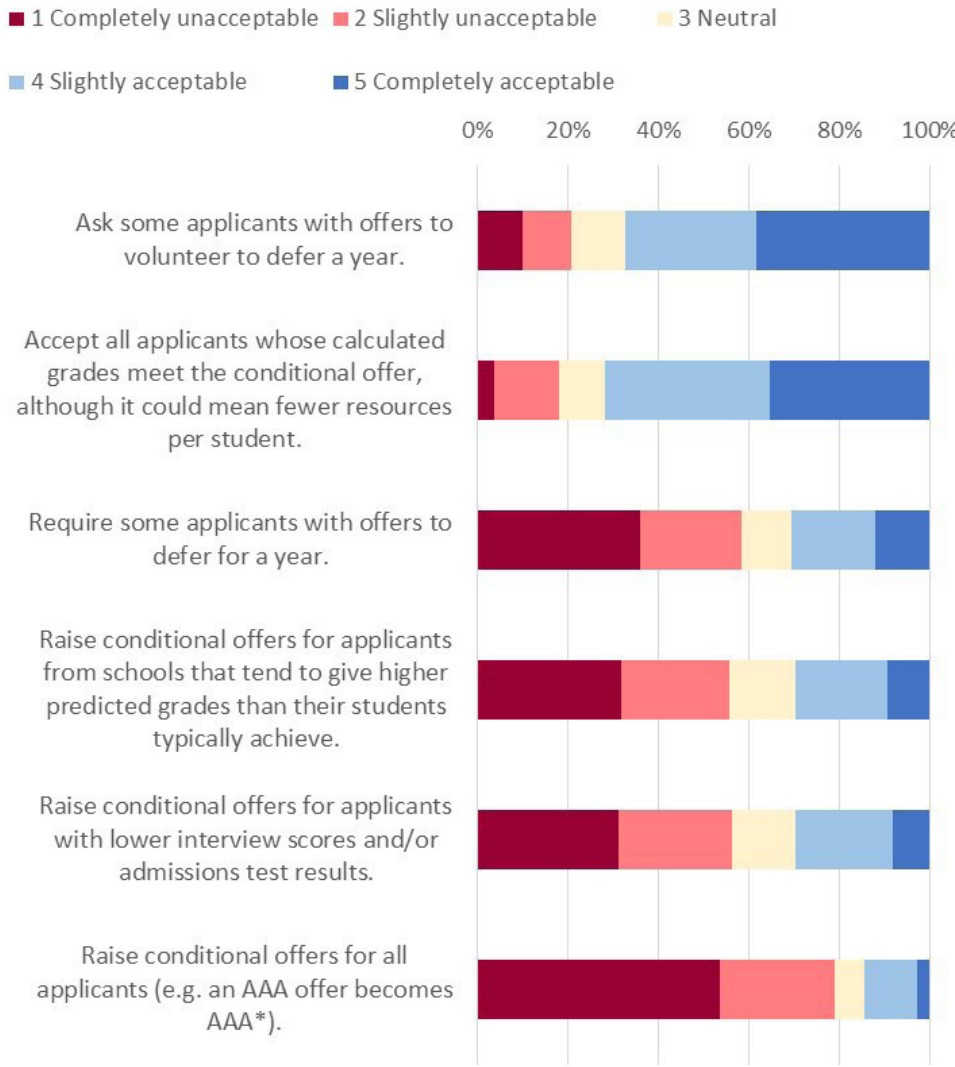

**Figure 3** Acceptability of actions medical schools could take if they have more applicants meeting offers than they have places.

NHS and allow to have more spaces. All these problems could be solved if exams were taken virtually.

The government could also provide more funding for medical schools- not only will this allow more people to attend but it will also mean there are more doctors down the line who can work in the NHS.

There were suggestions that applicants could opt to attend other medical schools they had applied to but which they had not selected as their firm or insurance choice, or that they could be offered places at medical schools they had not applied to:

If some medical schools have a lower numbers of applicants overall, compared with others, redistribute some students to these ones, with permission.

There were many suggestions of incentives to defer, and some felt that they would welcome a year off before starting:

Incentives to defer like 1 year free accommodation or £5000 or student ambassador job for gap year

Incentive to deferring such as free university accommodation for the first year, organised work experience placements and or organised care assistant jobs for the gap year.

If people are asked to volunteer to or forcefully defer entry, offering alternatives for work they could do within a healthcare setting for that year. For example, maybe clerical work within the NHS so they're still immersed within the healthcare system.

Asking students to voluntarily defer a year would be a popular option, I think many people will reevaluate their priorities over the coming months and may appreciate the opportunity.

The option to defer is definitely an option that should be considered as many people would be happy with the idea of gaining more medical experience in the year out that they would now have.

There were suggestions medical schools could have multiple cohorts either all starting in October or one cohort starting in October and another cohort starting early 2021.

Create an extra group/year for COVID-19 Students to manage the numbers

Maybe consider having staggered starts throughout the year October start January start June starts.

Stagger the course to offer two presentations and alter the following academic term holidays if possible

Respondents also expressed concern as to the impact of the present disruption on next year's admissions cycle and available resources:

The selection process should not be biased towards those rejected this year, next year, and should not change for the next cohort.

I hope that this year's or next year's applicants will not be disadvantaged due to these unprecedented circumstances.

### Perceptions of potential impact on admissions for 2021
Participants were asked to rate how much they agreed or disagreed with six options as to how medical schools could deal with the potential impact of the current situation on admissions in 2021 (see figure 4).

In general, respondents felt medical schools should give special consideration to current applicants re-applying next year (67.1% agreed/strongly agreed that *Applicants rejected this year should be given special consideration when re-applying next year*) however opinions were divided about what that special consideration should consist of.

Multiple regression analyses showed that after accounting for number of offers, educational, social and demographic factors, BAME respondents were more likely to feel that re-applicants should be given some advantages.

### Starting academic year 2020/2021
A majority of respondents (n=952, 61.1%) believed that if necessary, medical schools should *Defer the start of the academic year only when face-to-face teaching is possible* with 605 respondents (38.9%) believing that medical schools should *Start the academic year on time using distance learning for as long as is necessary*. This did not vary significantly by prior attainment, number of offers or educational/social/demographic background.

### Education and university preparation
### Calculated grades and the perceptions of process of awarding calculated grades in lieu of examination grades
Participants were generally fairly ambivalent towards calculated grades. On the positive side (see figure 5A), the majority of respondents (78.6%) preferred calculated grades to taking examinations next year, and about half (54.9%) preferred calculated grades to taking examinations in September 2020. Over half (59.3%) agreed that schools wouldn't be able to game the process to award all their students high grades, and 51.4% felt that the process of awarding calculated grades was the best way to be fair to most students in the circumstances (although 35.0% disagreed). Over half (56.4%) agreed that their teachers were generally able to rank and grade students accurately, however respondents were divided as to whether their own teachers knew them well enough to grade and rank them accurately: 42.0% agreed their teachers did *not* know them well enough whereas 44.6% thought their teachers *did* know them well enough.

On the negative side (see figure 5B), over half of respondents (52.9%) disagreed or strongly disagreed that calculated grades would result in an accurate assessment of their abilities, with 63.4% agreeing that teachers would find it hard to be unbiased, 80.7% agreeing it was difficult to see how teachers in large schools can rank so many students and 85.5% agreeing calculated grades

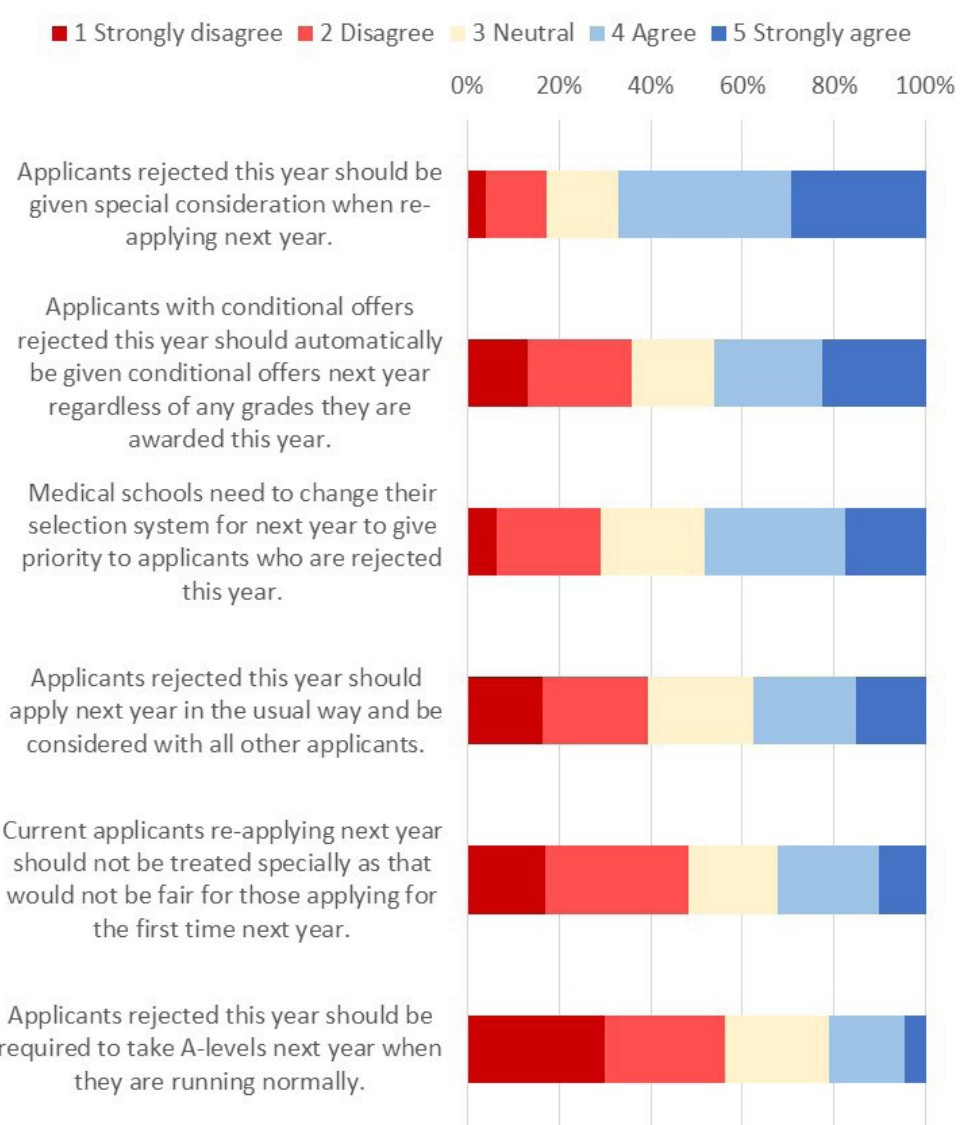

■ 1 Strongly disagree ■ 2 Disagree □ 3 Neutral ■ 4 Agree ■ 5 Strongly agree

**Figure 4** Views on how current applicants should be considered by medical schools if they reapply next year.

cannot take into account students doing better in exams than their teachers expected. Most agreed it was unfair to judge students on work done since schools/colleges closed (70.4%), that grades should be based solely on their performance and not the performance of previous students at their school (69.6%), and that it was unfair their GCSE performance was not taken into account (68.7%).

Mean top three predicted A-level points was a major predictor of perceptions of calculated grades but there were also differences by background after accounting for prior attainment, number of offers and other educational/social/demographic factors: BAME respondents and female respondents were more negative about calculated grades and respondents from non-selective state schools and those from more deprived areas were more likely to agree that calculated grades should not take into account the performance of previous pupils at their school (see table 2).

There were 398 freetext responses to the following request for further comments at the end of the questionnaire: 'Please use this space for any additional comments you wish to make about the questionnaire or selection of medical students'. These responses included concerns that calculated grades would be based on work completed early in the academic year and on mock exams created and assessed by the school. It was felt that these measures would not take into consideration the development and academic progress made by pupils over the year, even when teachers gave special consideration to the impact of the disruption. There was also concern that at the time of mock exams in particular, many medicine applicants were more focused on admission tests (BMAT in particular), submitting applications and preparing for interviews.

Grade calculations took away the chance the students had to prove themselves (final exams) and their control. Basing the final grade on a time when the

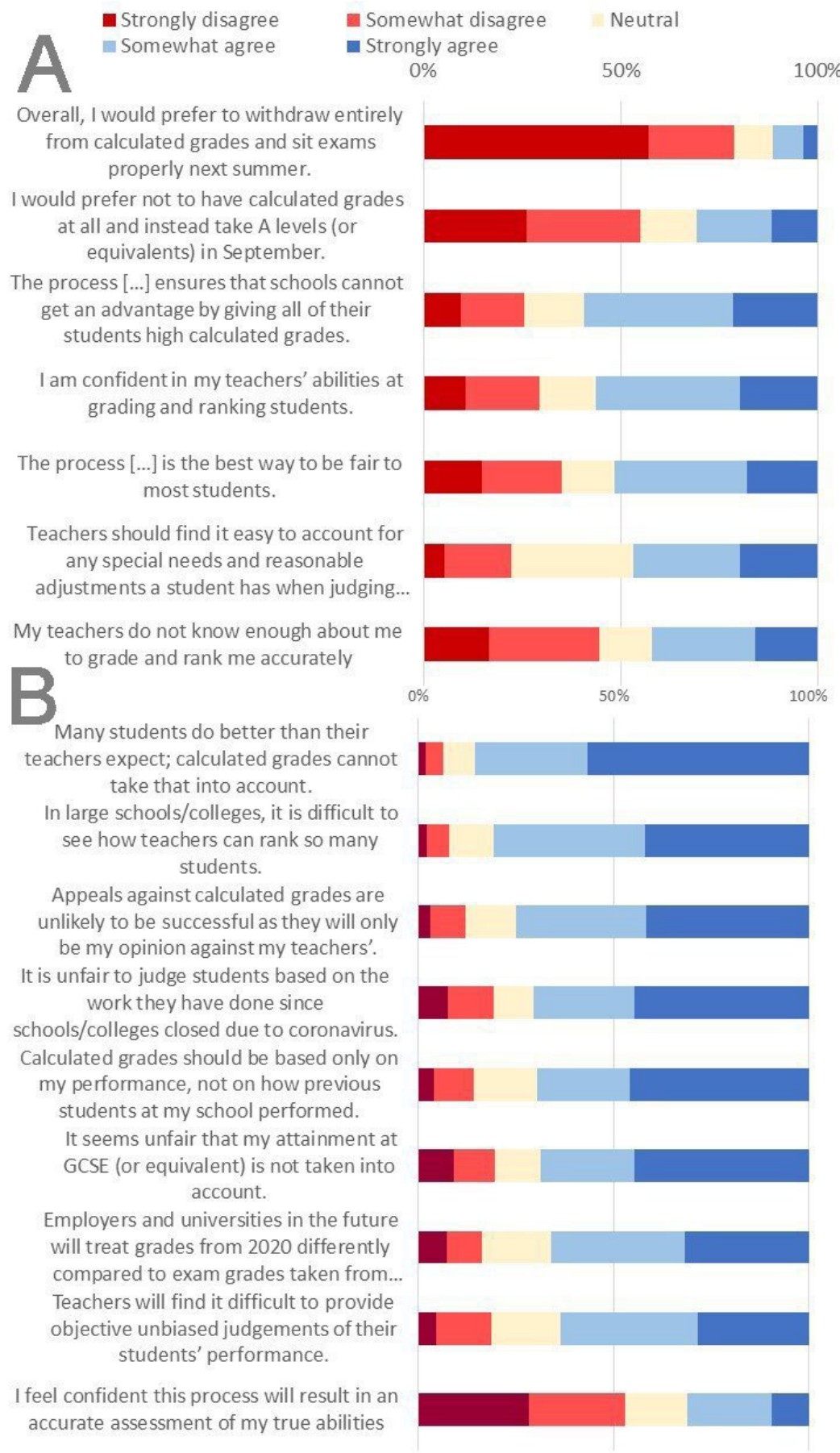

**Figure 5** Aspects of calculated grades that respondents were generally more (A) positive and (B) negative about.

**Table 2** Predictors of agreement with statements relating to calculated grades. Predictors are ordered left to right by strength of relationship to the statement. Only statements that showed significant differences by social/demographic group after controlling for prior attainment and the number of offers are shown

| | Independent predictors of agreement with statement | | | |
|---|---|---|---|---|
| I would prefer not to have calculated grades at all and instead take A levels (or equivalents) in September. | Lower predicted A-level points | BAME | Fewer conditional offers | Female |
| Overall, I would prefer to withdraw entirely from calculated grades and sit exams properly next summer. | Lower predicted A-level points | BAME | Fewer conditional offers | Female |
| The process described above is the best way to be fair to most students. | Higher predicted A-level points | White | Higher UCAT/BMAT scores | |
| I feel confident this process will result in an accurate assessment of my true abilities. | Higher predicted A-level points | White | Male | |
| Many students do better than their teachers expect; calculated grades cannot take that into account. | Lower predicted A-level points | BAME | | |
| My teachers should take into account the disruption caused by coronavirus when judging grades. | Lower predicted A-level points | BAME | | |
| Calculated grades should be based only on my performance, not on how previous students at my school performed. | Non-selective state school | Higher deprivation | | |
| I am confident in my teachers' abilities at grading and ranking students. | Higher predicted A-level points | White | | |
| My teachers do not know enough about me to grade and rank me accurately. | Lower predicted A-level points | BAME | Fewer conditional offers | |
| In large schools/colleges, it is difficult to see how teachers can rank so many students. | BAME | Lower predicted A-level points | | |
| Teachers judging grades should take into account the fact that many students do not do well in mocks but then work hard and do well in exams. | Lower predicted A-level points | Female | Fewer conditional offers | |
| Employers and universities in the future will treat grades from 2020 differently compared with exam grades taken from other years. | Female | Fewer conditional offers | | |

BAME, Black Asian and Minority Ethnic; UCAT, University Clinical Admissions Test.

students weren't aware that they were being truly assessed can hardly be classed as fair.

I believe universities should be lenient and realise that if a students calculated grade is below their conditional offer, this is not 100% representative of the students abilities. If they were able to secure an offer in the first place then universities should already know the academic capabilities of said student through their GCSE grades, predicted grades, UCAT/BMAT scores, teacher references, interviews etc. Otherwise, they wouldn't have given the student an offer. Where possible, every offer holder should be given their place at university in this academic year, whenever it resumes and should not be forced to take a year out and spend that year being stressed, lost and demotivated.

Teacher submitted grades being standardised by exam boards based on previous achievement from a school was a concern for this student:

I am the only student in my year and the third student in my sixth form's history to ever apply for medicine, and the first to receive all 5 offers. My school historically is one that does not do very well and I fear that my individual success and all the hard work I have had to do on my own as I get no help from my school, will be overshadowed by the bad results from previous years.

### Education since the shutdown

A minority of respondents said their school was planning on formally assessing them on work done since the shutdown (n=184; 11.8%); nearly half (n=740; 47.5%) said their school would not and over a third (n=614; 39.4%) were uncertain. Respondents attending a private/selective school were twice as likely to report being assessed on work since the shutdown (14.2% vs 7.6%; p<0.001).

Participants were asked whether they were using educational resources provided by their school/college and if not why not. Nearly all respondents had used at least one resource (n=1346; 91%) and three was the average number used.

Respondents attending private/selective schools were more likely to report having used all educational resources except support for university applications, and those at non-selective state schools used on average two resources compared with the three used by those at private/selective schools. The largest difference was in

**Table 3** School-provided educational resources used by respondents from non-selective state schools and private/selective schools

| | N (%) used resource | | | |
|---|---|---|---|---|
| | Non-selective state school | Private or selective school | Total | P value |
| Online resources | 342 (63.3) | 439 (80.0) | 781 (71.7) | <0.001 |
| Paper resources | 315 (58.3) | 375 (69.6) | 690 (63.9) | <0.001 |
| Online formative tests | 187 (34.8) | 260 (48.2) | 447 (41.5) | <0.001 |
| Pastoral support | 160 (29.7) | 199 (37.2) | 359 (33.4) | 0.009 |
| University application support | 152 (28.5) | 174 (32.3) | 326 (30.4) | 0.174 |
| Online teaching in real time | 66 (12.4) | 248 (45.7) | 314 (29.2) | <0.001 |
| Online summative tests | 70 (13.2) | 95 (17.7) | 165 (15.4) | 0.042 |
| Other | 12 (6.3) | 25 (14.2) | 37 (10.1) | 0.011 |

**Table 4** Respondents' main reasons for not using school educational resources during the shutdown by school type

| | | N (%) resource *not* used | | |
|---|---|---|---|---|
| Resource not used | Reason not used | Non-selective state school | Private or selective school | Total |
| Online resources | Not available | 96 (46.6) | 48 (43.2) | 144 (45.4) |
| | Don't need to | 80 (38.8) | 49 (44.1) | 129 (40.7) |
| Paper resources | Not available | 109 (50.5) | 74 (46.8) | 183 (48.9) |
| | Don't need to | 88 (40.7) | 69 (43.7) | 157 (42.0) |
| Online formative test | Not available | 206 (60.2) | 129 (48.3) | 335 (55.0) |
| | Don't need to | 119 (34.8) | 116 (43.3) | 235 (38.6) |
| Pastoral support | Not available | 161 (42.6) | 94 (28.4) | 255 (36.0) |
| | Don't need to | 194 (51.3) | 205 (61.9) | 399 (56.3) |
| University application support | Not available | 185 (49.9) | 141 (40.5) | 326 (45.3) |
| | Don't need to | 155 (41.8) | 182 (52.3) | 337 (46.9) |
| Online teaching in real time | Not available | 337 (71.7) | 189 (63.0) | 526 (68.3) |
| | Don't need to | 109 (23.3) | 99 (33.0) | 208 (27.0) |
| Online summative test | Not available | 289 (65.4) | 223 (52.5) | 512 (59.1) |
| | Don't need to | 142 (32.1) | 177 (41.6) | 319 (36.8) |
| Other | Not available | 66 (54.1) | 42 (39.3) | 108 (47.2) |
| | Don't need to | 47 (38.5) | 49 (45.8) | 96 (41.9) |

the use of online teaching in real time, which those at private/selective schools were nearly four times more likely to have used (see table 3).

In the multivariate analyses, attendance at a private/selective school was an independent predictor of using online teaching in real time, online resources for home learning, online formative assessments and paper resources for home learning, even after controlling for prior attainment and sociodemographics. In addition, having at least one parent/carer with a university degree was an independent predictor of using paper resources for home learning, and having lower UCAT/BMAT scores was an independent predictor of using online teaching in real time.

Those who had not used educational resources reported the main reason(s) were either that the resources were not available or that they felt they did not need to use them. Only very few said they had not used a resource because of a lack of private quiet space, lack of time, lack of internet/computer access or because they were finding it too hard to focus. Those at non-selective state schools were more likely than those at private/selective schools to state lack of availability as a reason, and less likely to state not needing to as a reason (see table 4).

### Preparation for medical school/university
Participants were asked what preparation if any they were doing for university or medical school (see figure 6).

Of the 207 (13.3% of the sample) who said they were not doing any preparation, the most common reason was that they were too worried and couldn't focus (n=88; 42.5% of those not doing any preparation), not having resources (35.5%), feeling it wasn't necessary (29.5%), caring for others (13.5%), not going to university this

year (14.0%), not having time (6.3%) and being unwell (4.8%). Respondents could select multiple reasons.

### Time spent during the lockdown
Participants were asked to state how much time they were spending on various activities in the previous 5 days (see figure 7). The multivariate analysis showed that respondents from private/selective schools reported spending more time studying, even after controlling for prior attainment and sociodemographic factors.

### Factor analysis
#### Number of factors
The factor analysis included 87 variables which are attitudinal or related to attitudes. The maximum eigenvalue was 6.99, with 27 eigenvalues greater than 1. A scree plot suggested that there was a break at or around six factors (see figure 8). Other criteria were very variable, with *fa.parallel()* in the *psych* package in *R* suggesting there were

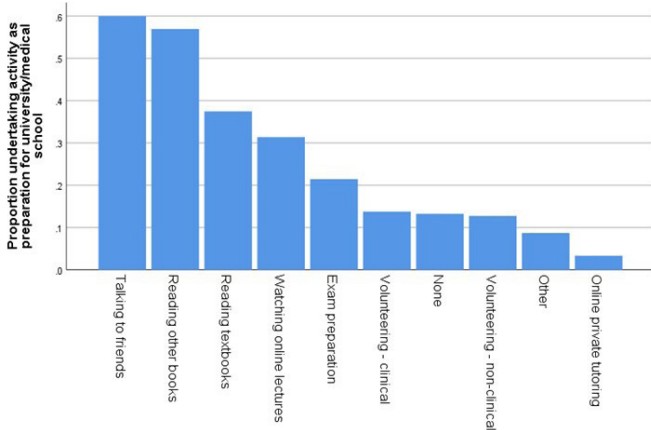

**Figure 6** Proportion of respondents undertaking various activities to prepare for medical school or university.

19 principal components with eigenvalues greater than the 95% upper CI for randomly generated data. *nfactors()* in *psych* said that VSS complexity 1 suggested 17 factors, VSS complexity 2 suggested 17 factors, Velicer's MAP gve 10 factors, Empirical BIC gave 20 factors and sample size adjusted BIC gave 20 factors. However the output also reported, 'Although the vss.max shows 17 factors, it is probably more reasonable to think about four factors'. Overall there are probably many small factors corresponding to measures with low communalities and hence mostly unique variance. For present purposes we are particularly interested in aggregating measures to gain more statistical power, and therefore we chose to extract six principal factors, which corresponds with the break in the scree slope, and is a little larger than the *nfactors()* recommendation of 4.

### Naming of factors

The six factors were named as following, by considering the highest absolute loadings, along with all loadings over 0.35:

1. '*Lack of confidence in calculated grades*'. Positive loadings (n=9 items) reflected concerns that teachers will not know students well enough and will find it hard to be objective, preferring not to have calculated grades and

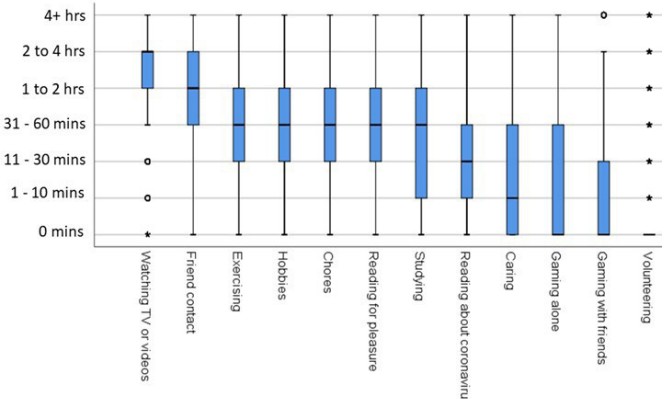

**Figure 7** Amount of time respondents reported spending on various activities during the lockdown.

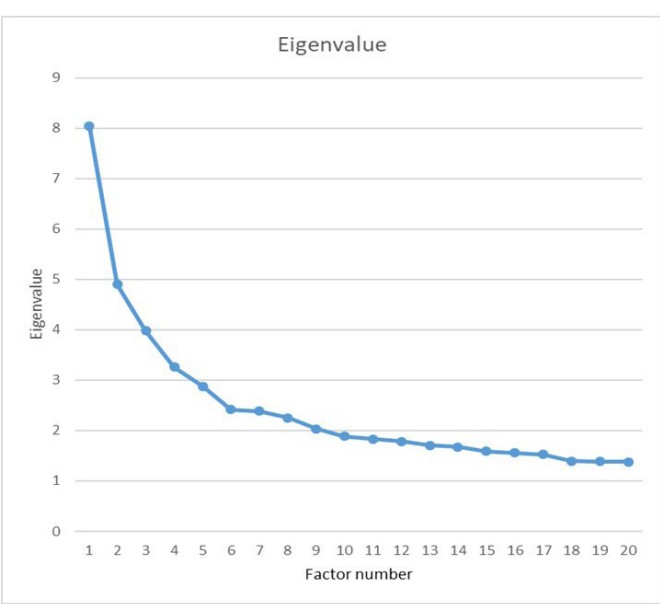

**Figure 8** Scree plot for the factor analysis of 87 attitudinal variables.

take exams in September or next summer, and appeals being unlikely to be successful. Negative loadings (n=5 items) reported confidence in the process resulting in an accurate reflection of a student's true ability, and the awarding process being fair to most students. High positive scores therefore represent a lack of confidence in the process of determining calculated grades.

2. '*Special treatment next year for rejected applicants*'. High positive loadings (n=6 items) were associated with medical schools needing to give higher priority and special consideration next year to students rejected this year, with rejected candidates being automatically given conditional offers next year. Negative loadings (n=4 items) suggested that re-applicants next year should be treated in the usual way, and special treatment for rejected applicants this year would not be fair for first year applicants next year. High positive scores therefore suggest that applicants who are rejected this year should be treated specially next year.

3. '*Other selection measures to be taken into account*'. A small group of items (n=3) suggested that selection could take into account aptitude tests such as UCAT, BMAT and performance at interviews. High scores therefore suggest that where possible, measures other than calculated grades should be taken into account.

4. '*Preparing for medical school*'. High positive loadings (n=4 items) reflected applicants who during lockdown were preparing for university by reading (either textbooks or other books), were watching online lectures, as well as talking with friends. Negative loadings (n=4 items) reflected applicants who were not doing any preparation, didn't feel preparation was necessary, didn't have any resources or who couldn't focus because they were too worried. High scores therefore indicate an applicant's concentration on preparing for medical school or university.

5. '*Importance of background and experience*'. All high loadings (n=8 items) were positive and indicated that medical schools should take into account work experience, the applicant's personal statement, and the teacher's reference on the UCAS form, attendance at university summer schools and widening participation programmes, an applicant's personal background such as being from under-represented groups, and other grades in qualifications such as GCSEs and the extended project qualifications. Overall higher scores indicate that a wider range of measures should be used to take into account personal background and wider experience.

6. '*Resources from school for home study*'. All loadings were positive (n=8 items), and indicated that applicants were being provided with live online teaching, online resources for home learning, paper resources such as workbooks, formative online assessments, and summative online assessments that might count towards calculated grades, doing timed essays or past papers, and spending more time studying. Higher scores therefore indicate having received greater support for home schooling from schools and colleges.

### Predictors of factor scores

Predictors of factor scores were assessed using multiple regression. All predictor variables in the set were entered and only those achieving $p < 0.01$ are reported. All predictors therefore take into account the effects of others in the set. Set A is the basic set used earlier in the study. Set B is extended by including socioeconomic group (based on parents' jobs), doctor parent(s) and the five Big Five personality factors, and are included on an exploratory basis (see table 5).

## SUMMARY AND CONCLUSIONS
### Summary of results

No single measure, including calculated grades, was considered fair enough by most applicants to use in the acceptance or rejection of offer holders; however many applicants considered calculated grades—and many other measures—fair enough to use in combination with other measures such as interview scores or admission test scores. Taking into account personal background or widening participation attendance was considered fairer by BAME applicants, those from deprived areas and those without degree-educated parents.

Many respondents had concerns about calculated grades, especially BAME and female applicants who felt teachers would find it difficult to grade and rank students accurately, and those from non-selective state schools and living in deprived areas were more concerned about the standardisation process that uses the attainment of previous pupils at a school. Despite this, the majority would rather have calculated grades than forgo calculated grades completely and take examinations in Autumn 2020 or Summer 2021 instead.

**Table 5** Predictors of factor scores. Set A includes Number of offers, GCSE points, Predicted A-level points, UCAT/BMAT score, Private/Selective school, Female, BAME, Degree-educated parent(s) and Deprived area. Set B includes Set A plus Highest socioeconomic group, doctor parent(s), and Big Five personality factors Agreeableness, Conscientiousness, Extraversion, Neuroticism and Openness. All predictors reported have $p < 0.01$, and are reported in descending order of significance (ie, most significant at the top)

| | Set A predictors in order of magnitude | Set B predictors in order of magnitude |
|---|---|---|
| Factor 1: Lack of confidence in calculated grades | Lower predicted A-levels | Lower predicted A-levels |
| | BAME | BAME |
| | Fewer conditional offers | Fewer conditional offers |
| | Female | Higher openness |
| | | Lower conscientiousness |
| | | Female |
| Factor 2: Special treatment next year for rejected applicants | Lower predicted A-levels | Lower predicted A-levels |
| | Lower UCAT/BMAT | Higher openness |
| | | Lower UCAT/BMAT |
| | | Higher neuroticism |
| | | Higher extraversion |
| Factor 3: Other selection measures to be taken into account | Higher UCAT/BMAT | Higher UCAT/BMAT |
| | Lower predicted A-levels | Lower predicted A-levels |
| | Male | Higher extraversion |
| | | Male |
| | | Lower conscientiousness |
| Factor 4: Preparing for medical school | White | Higher conscientiousness |
| | Female | Lower neuroticism |
| | | White |
| | | Female |
| | | Higher agreeableness |
| | | Higher openness |
| Factor 5: Importance of background and experience | Lower UCAT/BMAT | Higher openness |
| | BAME | Lower UCAT/BMAT |
| | Female | Fewer conditional offers |
| | | BAME |
| Factor 6: Resources from school for home study | Selective school | Selective school |
| | Lower GCSE | Lower GCSE |
| | Fewer conditional offers | Lower extraversion |
| | | Higher openness |

BAME, Black Asian and Minority Ethnic; UCAT, University Clinical Admissions Test .

Respondents mostly felt that medical schools should admit any applicant who met their conditional offer, even if that meant having to increase the number of places

(which would require a legal change and increased government funding), although there was also acceptance of medical schools asking for volunteers to defer but not of requiring deferrals. Respondents were divided as to how rejected applicants should be treated if they were to re-apply next year, with some respondents feeling they should be treated no differently and others feeling their 2020 experience should be taken into account. A majority of respondents tended to favour medical schools delaying the start of term until face-to-face teaching were possible.

Applicants from non-selective state schools reported using fewer educational resources than their counterparts at private or selective schools, and in particular they reported less online teaching in real time, and spending less time studying during the lockdown.

### Comparisons with other research

Our findings show many similarities to other recent UK studies of the effects of the COVID-19 pandemic on education in the UK;[8 9 15 16] however it is notable that in this sample of medical applicants ethnicity is more significant than socioeconomic factors in predicting concerns about calculated grades. A study of A-level students, conducted by Bhopal and Myers between April and August 2020 and published as a report on the OSF open access repository, surveyed an ethnically diverse sample of 583 A-level students in Britain and interviewed 53 students about their views on their education during the pandemic and their exam results. The authors report that 21% of students were glad exams had been cancelled but over twice as many (46%) would prefer to sit exams, which is similar to our finding that exams were considered the fairest method of selection. Similarly to our findings, the authors report that 'Many students also raised concerns their ethnicity could influence how teachers assessed their work' quoting a black student saying 'Some of my teachers seem biased […] They always think the Black boys are trouble', an Irish Traveller student saying 'We're Travellers. The school doesn't think much of us.' and an Indian student saying 'My teachers don't think I can do that well […] They also have their favourites, we can all see that – those students who they think should do well, are not those who necessarily will do well'. This reflects concerns from the BAME participants in our study about teacher bias.

It is known that predicted grades are lower for some minority ethnic groups[20] and indeed, on 2 April 2020 after the announcement of the cancellation of examinations but before *Ofqual* specified details of calculated grades, the Runnymede Trust and several other race equality organisations wrote to the Secretary of State for Education to urge him to 'ensure a fair, transparent and robust system which will more accurately reflect the ability and attainment of students from different backgrounds'.[21] Subsequently, on 30 April, the Equality and Human Rights Commission said that,

Using predicted grades in place of this year's summer assessments could deepen the existing inequality in education and put the future of disadvantaged young people at risk if not correctly implemented[22]

Our finding that students from private/selective schools were using more educational measures—especially online teaching in real time, which requires significant teacher input and which Andrew *et al*[15] argue is higher quality that other types of resource—reflects findings from those authors' research with parents of secondary school children[15] and teachers;[16] however in our sample students' use of educational resources and time spent studying did not vary by socioeconomic background, including parental higher education, socioeconomic status or area deprivation. This may be a feature of this particularly high-achieving sample of medical applicants.

### Strengths and limitations

This study is, to our knowledge, the first systematic exploration of medical applicant views on and experiences of the most significant changes to UK education in living memory. It is also the first study we are aware of that looked at university applicant views on calculated grades and the impact on university admissions. The large sample size gathered from around the UK, and the richness of the data allowed us to examine important differences in the experiences and views of different sociodemographic groups, after controlling statistically for educational attainment.

The speed at which we were required to develop the questionnaire and the unprecedented nature of the topic under investigation meant we were unable to use validated measures for most questions, nor have we been able to validate the measures ourselves, although we were able to pilot them with current applicants. Our data provide a snapshot of applicant views and experiences in April 2020, and it is possible that participants' views and experiences changed after data collection. The fact that participants are part of a longitudinal study however means we will have the chance to follow-up participants in 2021 and beyond to discover how the pandemic affected their education.

It is uncertain how representative our sample is of all medical applicants. Data on applications, offers, acceptances and academic achievement from the current UCAS cycle are not released until early 2021, but it is very likely that offer holders were over-represented in our sample. Data from the 2019 UCAT testing cycle also show that our sample scored higher than the mean (https://www.ucat.ac.uk/media/1329/2019-test-statistics-oct-2019.pdf); however not all UCAT test takers apply to medicine. Demographic data on 2020 medical applicants released by UCAS in November 2019 showed that our restricted sample was similar to all English applicants aged 17–19 years in terms of ethnicity and deprivation but had more women (https://www.ucas.com/data-and-analysis/undergraduate-statistics-and-reports/

ucas-undergraduate-releases/applicant-releases-2020/2020-cycle-applicant-figures-15-october-deadline).

Medical applicants are not representative of all university applicants in either academic or sociodemographic terms; however the similarity of some of our findings to that of other research, for example, that private school pupils are receiving significantly more education than non-selective state school pupils, suggests that the views and experiences of our sample may not be completely different from those of university applicants more generally; however generalisations from our findings to all applicants should only be done with caution.

## Implications for policy and practice

The impact of calculated grades on medical admissions was, at the time of writing, uncertain. Our questionnaire closed on 22 April and on 5 May 2020 the Medical Schools Council announced that medical schools would honour all offers met (something not clear at the time of our questionnaire), while acknowledging that there were still a number of issues that needed resolving.

How calculated grades are likely to work in practice has also been explored by a parallel analysis by our team using UKMED data over the last 10 years, comparing predicted A-level grades (which are likely to be similar to calculated grades) with actual, attained A-level grades.[23] Predicted grades were systematically higher in medical school applicants than eventually achieved grades. In addition, predicted grades predicted later outcomes only about two-thirds as well as achieved A-level grades predicted outcomes; this was true for outcomes five or six years later (at the end of medical school), and outcomes seven or eight years later (in postgraduate examinations). The underprediction by predicted grades was mitigated in part, although not entirely, by combining predicted grades with UCAT/BMAT scores, which supports the views of some applicants that other measures might be used for selection among applicants not meeting the terms of conditional offers.

The likely impacts on medical schools of using calculated grades were, at the time of writing, uncertain, but our estimates suggested there could in effect be a lowering of entry grade requirements, with possible subsequent increases in medical school dropout rates, and a somewhat academically weaker cohort with poorer performance in medical school and postgraduate examinations.[5 24] That is potentially important since very poor postgraduate examination performance itself strongly predicts being sanctioned by the medical regulator.[25]

In the awarding of calculated grades, we predicted that the raw 'centre assessment grades' and rankings produced by teachers for *Ofqual* were likely to be similar to predicted grades in being more generous than achieved A-level grades would have been, although the standardisation to be used by examination boards and *Ofqual* are likely to minimise that effect, so that distributions of calculated grades within subjects and centres become similar to actual A-level grades in previous years.

As it transpired the centre assessment grades ended up being used without adjustment, and these were significantly higher than previous years' A-level grades, with the Education Datalab stating 'At grades A*-A, there was an increase from 25.2% to 38.1%' (see https://ffteducation-datalab.org.uk/2020/08/gcse-and-a-level-results-2020-how-grades-have-changed-in-every-subject/).

As a result of the awarding of calculated grades an excess of candidates met their conditional offers (In the UK system, university offers are made before students take their exams. Universities typically give offers that are conditional on students achieving particular grades. Students meet their offer(s) and can be admitted if they achieve or exceed the grades specified.). Giving their views on what should happen in this regard, applicants in our study suggested that that in light of the shortage of doctors,[26] medical schools might argue for increased places and funding. In the event the Government did indeed lift the cap on medical school places to accommodate the increase in students (see https://www.gov.uk/government/news/action-agreed-to-support-students-into-preferred-universities). The impact of large increases in number on teaching and on predicting through to numbers of places for clinical teaching, foundation training and so on is still uncertain. It is worth considering that cohort sizes at many medical schools are already very large, that students tend to be less satisfied at larger schools,[27] and that accommodating extra students into face-to-face teaching that is COVID-secure is likely to be extremely challenging. On the other hand, there is a clear need for more doctors and it is likely that the change to admissions will result in a more socially and demographically diverse cohort.

In this questionnaire many applicants felt it could be fair to use other information such as interview score, UCAT score, or GCSE score to accept or reject offer holders, and this could include in selecting from among 'near-misses'. Overall respondents to our questionnaire demonstrate a lack of confidence in the process of calculated grades. Given the concerns of the Equality and Humans Rights Commission, and the clear concerns also expressed in our study by some disadvantaged groups, there is a clear need to ensure that entrants as far as possible continue to reflect the breadth of those applying to study medicine.

The cancellation of public examinations and the use of calculated grades are not the only problems facing the 2020 application cohort. They are also at risk, particularly those from non-selective state schools, of coming to medical school having had less education over the previous few months,[14] meaning medical schools may need to provide additional teaching and resources to help students catch up. This is likely to be especially challenging for medical schools given the huge constraints on university budgets arising from drops in student numbers[28] and given that many are likely to be unable to open for face-to-face teaching at the start of the academic year, which in itself has unknown consequences. The finding that BAME groups were more likely to think

teacher-estimated calculated grades could be unfair is concerning, and greater efforts need to be made to ensure education is fair and perceived as fair by students and all stakeholders. Transparent and independent analysis of the impact of exam cancellations on different sociodemographic groups, once data become available, will also be important.

The 2020 cohort of entrants is likely to face more uncertainty than any cohort of medical student entrants in the past half century, and our survey makes very visible the many concerns of those applicants.

## Conclusions

The global tragedy of the coronavirus pandemic, in addition to its extensive mortality and morbidity, has resulted in huge and sudden disruptions to established ways of life including education and training at all levels. Medical education and training is no exception. The coronavirus pandemic will have significant and long-term impacts on the selection, education and performance of our future medical workforce. Understanding how medical education will be affected is therefore important, and in particular how applicants - who are at the very start of their medical career - are being affected. Now more than ever we need medical education, and medical education research, to be prioritised and funded so we can ensure our future doctors are able to be resilient, successful and happy healthcare professionals providing excellent patient care. The present study provides a wide range of insights into the feelings of the 2020 cohort of applicants, only a small proportion of which we have adequately been able to report here, but which demonstrate the concerns many have about examination cancellations in 2020 and looking forward to 2021

**Acknowledgements** Firstly we are immensely grateful to the several thousand medical school applicants who took the time to respond to survey with a very tight time window, and we particularly thank those who commented that they were pleased that the survey they gave them an opportunity to express their thoughts, feelings and anxieties. We could not include everything that was said, but all comments have been read by the team, and we hope that the current paper summarises some of those many and varied views. We are also grateful to Paul Garrud, Clare Owen, Konstantinos Lulo, and Ewan McNicol for their comments on earlier versions of the questionnaire.

**Contributors** KW, DH and CM jointly developed the idea for the study, and developed the questionnaire together. DH was responsible for putting the questionnaire online, and for identifying applicants to whom it should be sent, as well as sending text and email reminders. DH and KW cleaned the data, and KW, DH and CM were all involved in data analysis. The report was written jointly by all three authors, and all authors have read and reviewed the final draft.

**Funding** KW is a National Institute for Health Research (NIHR) Career Development Fellow (NIHR CDF-2017-10-008) and is principal investigator for the UKMACS and UKMEDP089 projects supported by the NIHR funding. DH is funded by NIHR grant CDF-2017-10-008 to KW. CM has received no specific funding for this project. This publication presents independent research funded by the NIHR. The views expressed are those of the author(s) and not necessarily those of the NHS, the NIHR or the Department of Health and Social Care.

**Disclaimer** KW and DH state that this publication presents independent research funded by the National Institute for Health Research (NIHR). The views expressed are those of the authors and not necessarily those of the NHS, the NIHR or the Department of Health and Social Care.

**Competing interests** All authors have completed the ICMJE uniform disclosure form at http://www.icmje.org/coi_disclosure.pdf: KW and DH report grants and non-financial support from the National Institute for Health Research during the conduct of the study; and KW reports personal fees from Transforming Student Access and Outcomes (TASO), outside the submitted work. All authors report no other relationships or activities that could appear to have influenced the submitted work.

**Patient consent for publication** Not required.

**Ethics approval** The study was approved by the UCL Research Ethics Committee Chair on 8 April 2020 as an amendment to the ongoing UKMACS longitudinal questionnaire study (reference: 0511/014).

**Provenance and peer review** Not commissioned; externally peer reviewed.

**Data availability statement** Data are available upon reasonable request. The data will be linked into the UK Medical Education Database www.ukmed.ac.uk to which researchers can apply for access.

**ORCID iD**
Katherine Woolf http://orcid.org/0000-0003-4915-0715

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
