## [Reviewer comments · BMJ Open]

ARTICLE DETAILS

TITLE (PROVISIONAL)	The attitudes, perceptions and experiences of medical school applicants following the closure of schools and cancellation of public examinations in 2020 due to the COVID-19 pandemic: a cross-sectional questionnaire study of UK medical applicants
AUTHORS	Woolf, Katherine; Harrison, David; McManus, Chris

VERSION 1 – REVIEW

REVIEWER	John Ryan University of Utah
REVIEW RETURNED	20-Sep-2020

GENERAL COMMENTS	In this original manuscript by Woolf and colleagues, the authors study the effect of closure of schools/universities/exams due to COVID-19 on medical applicants in the UK. The authors surveyed 2887 participants who had registered to take the University Clinical Admissions Test (UCAT) and who had been considering applying to medicine in the UK. The authors found that there was concern about the calculated grades that replaced the A-level results – in particular, the respondents were concerned that female, Black Asian and Minority Ethnic (BAME) applicants would not be graded accurately. This is a potentially important study because it highlights the impact of COVID-19 on the medical workforce. If the concerns are accurate, with the cancellation of exams and closures of schools etc will have resulted in women, BAME and other socio-economically disadvantaged groups being recruited less for medical schools. Based on this study, there is concern about how systemic racism will manifest itself in medical school admissions after the COVID-19 pandemic. This study is limited by lack of follow up survey data. Major concerns The concerns raised by the students reflects concerns about systemic and institutional racism. However, the authors never directly comment on racism or verify that the students have concerns about racism. Instead they talk about bias. Why is that? Can the authors comment on the availability of follow up data as this data is already 5 months in a fast moving pandemic. Is there data on how this has impacted other areas, for example business schools?
--

	Can the authors give suggested recommendations as to how to solve the perceptions that students have? Much of the information in the introduction could be changed to the discussion (it takes a while to get to the meat of the study) Many of Ofqual's recommendations regarding grades and admissions were made after the survey was completed. Can the authors explain why they used the two week window that was chosen (April 8th to 22nd) and how that might have influenced the validity their results? What % of UKMACS participants were ultimately included and excluded? It would be valuable to have a traditional Figure 1 with arrows showing who was being excluded etc How was the questionnaire validated? How do the authors verify that the 15% that responded are representative. It is a pity that ethnicity was missing in 43% of respondents. "Participants self-reported their current or most recent school in the current questionnaire" How did the authors confirm the accuracy of this? "Respondents who did not have a parent/carer with a university degree were less likely to have a medical offer" Were there any other variables seen that impacted offers? "In general, respondents felt medical schools should give special consideration to current applicants reapplying next year" This is self serving. One of the most concerning features is that 13.3% were not doing any preparation for medical school. The authors state: "It is generally agreed that there are five distinct personal traits or factors". I am not sure this is true. I would caution about making inferences on personality traits (however many you want to list) based on a non-validated survey
--	---

REVIEWER	Roger Ruiz Moral School of Medicine. Universidad Francisco de Vitoria. Madrid. Spain
REVIEW RETURNED	15-Oct-2020

GENERAL COMMENTS	"The attitudes, perceptions and experiences of medical school applicants following the closure of schools and cancellation of public examinations in 2020 due to the COVID-19 pandemic" I appreciate the opportunity that BMJ Open offers me to review this interesting work whose objective is to describe the experiences and views of medical applicants from diverse social backgrounds following the closure of schools and universities and the cancellation of public examinations in the United Kingdom (UK) due to COVID-19 / coronavirus.
---

	I congratulate the authors because the study represents an important and valuable effort to obtain the vision of medical applicants in the face of the extraordinary situation of school closings and cancellation of public examinations and thus obtain unpublished perspectives on important and sensitive issues to consider in a situation that, even today, has not been solved and that unfortunately and probably will not be solved quickly in future months. This further highlights the value of their contributions. Of particular interest is the emphasis given on obtaining the opinions of ethnic minorities, which is particularly important when addressing measures that need contemplate and minimize social inequality when access to a medical school. I believe that the focus and methodological development of the study is adequate in all its facets, it is exhaustive and well detailed. The study extends its analysis in a broad way on different independent variables and their relationship with the variables studied and includes the qualitative perspective through comments from the respondents. Nevertheless, and in contrast to the quantitative approach, no details are offered on the methodological treatment of this qualitative perspective of the study. Anyway, this represents a minor aspect. Although it is not the objective of this study, its conclusions would highlight the interest of also knowing the perspective that teachers have on some of the issues raised here. My main concern regarding this study in its current presentation, however, is related to the suitability of its publication in a journal such as BMJ Open. In my view, it is precisely that level of detail and depth of analysis in the broad description of the attitudes, perceptions and experiences of medical school applicants, which may make it inappropriate, since this presentation of the study would represent more a kind of “dossier” or “report” and would not fit well to the standards recommended by BMJ Open. Certainly the presentation of the work in 6500 words (roughly), 13 figures and tables plus two supplementary files with other additional 14 figures and tables, will surely impact upon the paper's 'readability' by the average BMJ Open reader. Having said that, I would highlight among the main conclusions of this work the concern of the applicants respondents about the fact that the calculated grades were as generally not considered fair enough to use in selection, but were considered fair enough to use in combination with other measures including interview and aptitude test scores, and the interesting fact that female and Black Asian and Minority Ethnic (BAME) applicants felt teachers would find it difficult to grade and rank students accurately. I would then suggest to the authors that they select both the analyzes performed and the results obtained so that they respond as understandably and adequately as possible supporting those conclusions and better adapting to the BMJ Open recommendations for the authors. In any case, I congratulate to the authors for this work.
REVIEWER	Donald Munro Conjoint Assoc Prof School of Psychology University of Newcastle NSW 2308 Australia

	I have met one of the authors, Prof. Chris McManus, at a conference several years ago
REVIEW RETURNED	30-Nov-2020

GENERAL COMMENTS	REVIEW of MS bmjopen-2020-044753 "The attitudes, perceptions and experiences of medical school applicants following the closure of schools and cancellation of public examinations in 2020 due to the COVID-19 pandemic" Received November 2020 Date of review submission 30TH November 2020-11-30 Recommendation: Publish speedily, subject to minor editing (and possible abbreviation). General overview: The article is unusual in the medical education field both in respect of the unique circumstances of its genesis and in the research methodology, which was a social survey carried out urgently. It provides an important contribution to our understanding of the attitudes of potential medical students to the extraordinary conditions imposed by the Covid19 pandemic and by government and administrative reactions to that pandemic. While there may be no repeat of these circumstances in the longer term, the findings will be of considerable use to medical school administrators in the near future, in UK as well as those countries with a similar way of selecting medical students. The findings should be made available to all such medical schools at an early date, in the form of a pre-publication if there is any delay in journal publication. I also believe that a follow-up of the participants during their medical school career and beyond would be of value. In general terms the article is well written and theoretically coherent, the methodology is appropriate and the results are clearly presented. However, it is quite long, and its impact might possibly be increased by some reduction, at least for immediate publication. Some suggestions are given below. Detailed notes (using page numbers at top of page; page/line references) 4/23 to 7/9 This section is clear and interesting, though possibly some of the details could be removed to appendices, leaving only the gist of the developments leading up to the survey. On the other hand, if any studies of UK (or other) medical school applicants in "normal times" have been published they might be briefly mentioned as previous research (with any qualitatively significant differences in findings given in the Results or Discussion sections).
--

	Possibly include references at 19/6 here? See also comment 8/10 below./span> 4/27 Specify October 2020 for foreign readers? 4/56 It may be useful to cite a web page for Ofqual 4/58 Estimated grades – what is publically known or believed about their reliability and validity in relation to examination scores? Also, what kind of estimation are teachers asked for (specific score, range, etc)? 6/20 Here and elsewhere there are problems with the tense of verbs, due to the unknown date of publication, so some sentences here and elsewhere may have to be reviewed when that is known. 7/31 not clear what the criteria for “seriously considering” were. 7/34 invite > invitation? 7/51 word(s) missing? 8/10 reference(s) to ‘previous UKMACS questionnaires’? 8/39-8/59 Reduce/remove technical detail? 8/54 For some variables (e.g., ethnicity, school type) the number of available categories would be useful (refer to Table 1?) 9/36-59 Consider removing to appendix (referenced for Missing Data)? 10/19 “over an A grade” not clear 11/27-46 and elsewhere Consider brief extracts that provide useful suggestions or comments 14/44 to game 17/28-18/16 Give number of items in each chosen factor? 18/15 Specify “having received greater support . . .” to avoid misinterpretation as a need for greater support? 20/23 Sentence beginning “In addition . . .” is difficult to understand 20/53 ‘met’ not clear 31 Table 5 I would suggest deleting this table and references to it as the results are not relevant to the other findings (but it would certainly be worth a separate publication in a psychology journal or as a conference presentation)
--	---

VERSION 1 – AUTHOR RESPONSE

Reviewer: 1

Reviewer Name: John Ryan

Reviewer: 2

Reviewer Name: Roger Ruiz Moral

Reviewer: 3

Reviewer Name: Donald Munro

Reviewer: 1

Institution and Country: University of Utah

Reviewer: 2

Institution and Country: School of Medicine. Universidad Francisco de Vitoria. Madrid. Spain

Reviewer: 3

Institution and Country: Conjoint Assoc Prof School of Psychology University of Newcastle NSW 2308 Australia

Reviewer: 1

Comments to the Author

In this original manuscript by Woolf and colleagues, the authors study the effect of closure of schools/universities/exams due to COVID-19 on medical applicants in the UK. The authors surveyed 2887 participants who had registered to take the University Clinical Admissions Test (UCAT) and who had been considering applying to medicine in the UK. The authors found that there was concern about the calculated grades that replaced the A-level results – in particular, the respondents were concerned that female, Black Asian and Minority Ethnic (BAME) applicants would not be graded accurately.

This is a potentially important study because it highlights the impact of COVID-19 on the medical workforce. If the concerns are accurate, with the cancellation of exams and closures of schools etc will have resulted in women, BAME and other socio-economically disadvantaged groups being recruited less for medical schools. Based on this study, there is concern about how systemic racism will manifest itself in medical school admissions after the COVID-19 pandemic.

This study is limited by lack of follow up survey data.

We agree that follow-up data will be important. They are not currently available for linkage with our cohort data, but once they are we intend to analyse them. An important feature to note is that the data of UKMACS are being integrated into the UKMED dataset, and hence follow-up, using administrative data on progression etc, will be available in perpetuity. We have now emphasised that in the report.

Major concerns

The concerns raised by the students reflects concerns about systemic and institutional racism. However, the authors never directly comment on racism or verify that the students have concerns about racism. Instead they talk about bias. Why is that?

This is an interesting question, and we thank the reviewer for raising it. So far as we are aware, we have not used the word bias to discuss the findings of our study. We quote several participants who use the word (e.g. “calculated grades which have the potential to be biased.” ... “there will definitely be a bias in terms of how certain students achieved their grade”). One of our survey questions asked whether “teachers would find it hard to be unbiased” in awarding grades. Here we didn’t specifically mention race or ethnicity, because we wanted to ask about potential unfairness in relation to whichever social, demographic or other characteristics (e.g. class, gender etc) participants felt was most relevant.

Can the authors comment on the availability of follow up data as this data is already 5 months in a fast moving pandemic.

We agree this is an important point. As stated on p18 in the Discussion, follow-up data on UK medical applications and acceptances will be available within the UK Medical Education Database (UKMED) in 2021. We are submitting these questionnaire data for linkage within UKMED and we intend to analyse those data in due course. We have included a bullet point in the “Strengths and limitations” section to reflect the importance of follow-up data.

Is there data on how this has impacted other areas, for example business schools?

At the time of writing we were not aware of any other similar studies of A-level students. However a study of A-level students, conducted by Bhopal and Myers between April and August 2020, has now been published as a report on the OSF open access repository, which we have now added to the discussion.

Can the authors give suggested recommendations as to how to solve the perceptions that students have?

Thank you for suggesting we include this. We have now added the following two sentences to the discussion:

“The finding that Black Asian and Minority Ethnic groups were more likely to think teacher-estimated calculated grades could be unfair is concerning, and greater efforts need to be made to ensure education is fair and perceived as fair by students and all stakeholders. Transparent and independent analysis of the impact of exam cancellations on different sociodemographic groups, once data become available, will also be important.”

Much of the information in the introduction could be changed to the discussion (it takes a while to get to the meat of the study)

As also suggested by other reviewers, we have cut down the Introduction and moved some information to the Supplementary Material.

Many of Ofqual’s recommendations regarding grades and admissions were made after the survey was completed. Can the authors explain why they used the two week window that was chosen (April 8th to 22nd) and how that might have influenced the validity their results?

We intended to use the study results to inform medical school admissions staff about applicant views and experiences. We knew that medical school staff were meeting in May 2020 and therefore we needed to collect and analyse data in time to provide staff with information by that date. In a fast-moving pandemic it was only possible for us to gather a snapshot of data, and of course it is likely that some or many participants’ views and experiences changed after data collection. We did however

have the advantage of longitudinal data from earlier survey(s) and the possibility of following up participants (as discussed above), which provides information beyond just a snapshot.

We have added this sentence into the Strengths and Limitations section of the Discussion: “Our data provide a snapshot of applicant views and experiences in April 2020, and it is likely that participants’ views and experiences changed after data collection. The fact that participants are part of a longitudinal study however means we will have the chance to follow up participants in 2021 and beyond to discover how the pandemic affected their education.”

What % of UKMACS participants were ultimately included and excluded? It would be valuable to have a traditional Figure 1 with arrows showing who was being excluded etc

Thank you we have now included this.

How was the questionnaire validated?

We were not able to validate the survey in its entirety, but as explained on p7: “Most questions were designed specifically for this questionnaire since they asked about unprecedented events and validated items were not available. We constructed the questionnaire with JISC Online Surveys [<https://www.onlinesurveys.ac.uk/>] and piloted the questionnaire and information sheet with two current applicants to medical school. Amendments were made in response to feedback from the applicants and from Medical Schools Council.”

How do the authors verify that the 15% that responded are representative.

We are not currently able to be certain how representative they are. As explained on p18: “It is uncertain how representative our sample is of all medical applicants. Data on applications, offers, acceptances and academic achievement from the current UCAS cycle are not released until early 2021, but it is very likely that offer-holders were over-represented in our sample. Data from the 2019 UCAT testing cycle also show that our sample scored higher than the mean [<https://www.ucat.ac.uk/media/1329/2019-test-statistics-oct-2019.pdf>]; however not all UCAT test-takers apply to medicine. Demographic data on 2020 medical applicants released by UCAS in November 2019 showed that our restricted sample was similar to all English applicants aged 17 to 19 in terms of ethnicity and deprivation but had more women [<https://www.ucas.com/data-and-analysis/undergraduate-statistics-and-reports/ucas-undergraduate-releases/applicant-releases-2020/2020-cycle-applicant-figures-15-october-deadline>].”

It is a pity that ethnicity was missing in 43% of respondents.

We agree. On reflection, we would have asked participants their ethnicity in the current questionnaire instead of relying on an earlier Wave of the questionnaire to collect these data.

“Participants self-reported their current or most recent school in the current questionnaire” How did the authors confirm the accuracy of this?

We were unable to confirm the accuracy of this with the current data. This is why we are clear to state that school was self-reported.

“Respondents who did not have a parent/carer with a university degree were less likely to have a medical offer” Were there any other variables seen that impacted offers?

Yes, we have added these in.

“In general, respondents felt medical schools should give special consideration to current applicants reapplying next year” This is self serving.

We are clear that this study is reporting participants’ views. These can be interpreted in various ways.

One of the most concerning features is that 13.3% were not doing any preparation for medical school.

15% of participants were not holding a medical school offer, and many participants reported in previous surveys that if they were not successful they would re-apply the following year. It may be that’s what those who weren’t preparing were planning on doing, and in due course (by 2022) we will be able to find out from follow-up administrative data in UKMED.

The authors state: “It is generally agreed that there are five distinct personal traits or factors”. I am not sure this is true. I would caution about making inferences on personality traits (however many you want to list) based on a non-validated survey.

The Big 5 personality items were in fact based on a validated survey from the national long-running longitudinal survey Understanding Society, as detailed on p7 and with reference 17.

Reviewer: 2

Comments to the Author

“The attitudes, perceptions and experiences of medical school applicants following the closure of schools and cancellation of public examinations in 2020 due to the COVID-19 pandemic”

I appreciate the opportunity that BMJ Open offers me to review this interesting work whose objective is to describe the experiences and views of medical applicants from diverse social backgrounds following the closure of schools and universities and the cancellation of public examinations in the United Kingdom (UK) due to COVID-19 / coronavirus.

I congratulate the authors because the study represents an important and valuable effort to obtain the vision of medical applicants in the face of the extraordinary situation of school closings and cancellation of public examinations and thus obtain unpublished perspectives on important and sensitive issues to consider in a situation that, even today, has not been solved and that unfortunately and probably will not be solved quickly in future months. This further highlights the value of their contributions.

Of particular interest is the emphasis given on obtaining the opinions of ethnic minorities, which is particularly important when addressing measures that need contemplate and minimize social inequality when access to a medical school. I believe that the focus and methodological development of the study is adequate in all its facets, it is exhaustive and well detailed. The study extends its analysis in a broad way on different independent variables and their relationship with the variables studied and includes the qualitative perspective through comments from the respondents.

Thank you for your kind comments.

Nevertheless, and in contrast to the quantitative approach, no details are offered on the methodological treatment of this qualitative perspective of the study. Anyway, this represents a minor aspect.

The focus of this study was quantitative, and as described on p7, we used quotes to illustrate rather than performing a full analysis of the qualitative data.

Although it is not the objective of this study, its conclusions would highlight the interest of also knowing the perspective that teachers have on some of the issues raised here.

We agree that this would be interesting. Reference 16 is for TeacherTapp, an ongoing survey of teachers, which does provide data on teacher perceptions of education during the pandemic.

My main concern regarding this study in its current presentation, however, is related to the suitability of its publication in a journal such as BMJ Open. In my view, it is precisely that level of detail and depth of analysis in the broad description of the attitudes, perceptions and experiences of medical school applicants, which may make it inappropriate, since this presentation of the study would represent more a kind of "dossier" or "report" and would not fit well to the standards recommended by BMJ Open. Certainly the presentation of the work in 6500 words (roughly), 13 figures and tables plus two supplementary files with other additional 14 figures and tables, will surely impact upon the paper's 'readability' by the average BMJ Open reader.

Having said that, I would highlight among the main conclusions of this work the concern of the applicants respondents about the fact that the calculated grades were as generally not considered fair enough to use in selection, but were considered fair enough to use in combination with other measures including interview and aptitude test scores, and the interesting fact that female and Black Asian and Minority Ethnic (BAME) applicants felt teachers would find it difficult to grade and rank students accurately. I would then suggest to the authors that they select both the analyzes performed and the results obtained so that they respond as understandably and adequately as possible supporting those conclusions and better adapting to the BMJ Open recommendations for the authors.

As suggested by other reviewers, we have shortened the manuscript and added more information into an Appendix.

In any case, I congratulate to the authors for this work.

Many thanks, we appreciate your comments.

Reviewer: 3

Comments to the Author

REVIEW of MS bmjopen-2020-044753 "The attitudes, perceptions and experiences of medical school applicants following the closure of schools and cancellation of public examinations in 2020 due to the COVID-19 pandemic"

Received November 2020 Date of review submission 30TH November 2020-11-30

Recommendation: Publish speedily, subject to minor editing (and possible abbreviation).

General overview: The article is unusual in the medical education field both in respect of the unique circumstances of its genesis and in the research methodology, which was a social survey carried out urgently. It provides an important contribution to our understanding of the attitudes of potential medical students to the extraordinary conditions imposed by the Covid19 pandemic and by government and administrative reactions to that pandemic.

While there may be no repeat of these circumstances in the longer term, the findings will be of considerable use to medical school administrators in the near future, in UK as well as those

countries with a similar way of selecting medical students. The findings should be made available to all such medical schools at an early date, in the form of a pre-publication if there is any delay in journal publication. I also believe that a follow-up of the participants during their medical school career and beyond would be of value.

We agree that follow-up will be important, and have added this in to the Strengths and Limitations section, as explained in response to Reviewer 1. In particular, through UKMED we will be able to follow up entrants for the whole of their medical education career, and so in the future will be able to assess the long-term impact of COVID on medical school admissions.

In general terms the article is well written and theoretically coherent, the methodology is appropriate and the results are clearly presented.

Thank you.

However, it is quite long, and its impact might possibly be increased by some reduction, at least for immediate publication. Some suggestions are given below.

Thank you, we have responded to each suggestion in turn.

Detailed notes (using page numbers at top of page; page/line references)

4/23 to 7/9 This section is clear and interesting, though possibly some of the details could be removed to appendices, leaving only the gist of the developments leading up to the survey. On the other hand, if any studies of UK (or other) medical school applicants in “normal times” have been published they might be briefly mentioned as previous research (with any qualitatively significant differences in findings given in the Results or Discussion sections). Possibly include references at 19/6 here? See also comment 8/10 below.

We have streamlined this section as much as possible. We are not aware of any other surveys that have asked medical applicants similar questions because the focus of the questionnaire was very much related to experiences during the pandemic (eg. changes to medical school admissions, views on calculated grades, experiences of teaching during school shutdowns).

4/27 Specify October 2020 for foreign readers?

Unfortunately we are unsure what this refers to.

4/56 It may be useful to cite a web page for Ofqual

We have added this in to the reference.

4/58 Estimated grades – what is publically known or believed about their reliability and validity in relation to examination scores? Also, what kind of estimation are teachers asked for (specific score, range, etc)?

Calculated grades were a new invention in light of COVID, so it’s uncertain what was publically known or believed about their reliability or validity, other than what we have reported in the paper. We have added in some more information about what teachers were asked to estimate both in the text and in a footnote, and have referenced a paper we wrote on calculated grades at the time published on MedRXiv and also submitted for publication in a peer reviewed journal.

6/20 Here and elsewhere there are problems with the tense of verbs, due to the unknown date of publication, so some sentences here and elsewhere may have to be reviewed when that is known.

Thank you. We have changed some of the tenses to reflect the fact that students have now been awarded their grades.

7/31 not clear what the criteria for “seriously considering” were.

We have now added in a footnote to explain this.

7/34 invite > invitation?

We have changed “invite” to “invitation” on p6.

7/51 word(s) missing?

Unfortunately we are not sure what this refers to.

8/10 reference(s) to ‘previous UKMACS questionnaires’?

On p6 we explain that UKMACS is a longitudinal questionnaire study, and talk about previous questionnaires. We have now added in a footnote on p6 to explain when Wave 1 and Wave 2 data were collected.

8/39-8/59 Reduce/remove technical detail?

We have this into a “Missing data” section in the Supplementary file.

8/54 For some variables (e.g., ethnicity, school type) the number of available categories would be useful (refer to Table 1?)

We have added a clause to the first sentence under the Demographics heading, to signpost that details of categories within variables can be found in Table 1.

9/36-59 Consider removing to appendix (referenced for Missing Data)?

We have moved most of the missing data section into the Supplementary file.

10/19 “over an A grade” not clear

We have removed this.

11/27-46 and elsewhere Consider brief extracts that provide useful suggestions or comments

We have cut down a quote to focus on suggestions, but have kept in where participants talk about perceived bias in assessments, because that is relevant to the main study findings.

14/44 to game

Amended, thank you.

17/28-18/16 Give number of items in each chosen factor?

We have added in the number of items loading onto each factor.

18/15 Specify "having received greater support . . ." to avoid misinterpretation as a need for greater support?

Amended, thank you.

20/23 Sentence beginning "In addition . . ." is difficult to understand

We have reworded this to hopefully make it clearer.

20/53 'met' not clear

In the UK system, university offers are made before students take their exams. Universities typically give offers that are conditional upon students achieving particular grades. Students meet their offer(s) if they achieve or exceed the grades specified. We have included this explanation as a footnote.

31 Table 5 I would suggest deleting this table and references to it as the results are not relevant to the other findings (but it would certainly be worth a separate publication in a psychology journal or as a conference presentation)

We have done as suggested.

As I have said in the full comments, this is an important paper that should be shared with medical schools in UK and elsewhere in a timely fashion. I have made a few suggestions for improvements and corrections, among them the possibility of some reduction in length, but the paper is generally interesting and valuable to the medical education community.

Many thanks indeed.

Reviewer: 1

Competing interests 1: None

Reviewer: 2

Competing interests 1: None declared

Reviewer: 3

Competing interests 1: I have met one of the authors, Prof. Chris McManus, at a conference several years ago

VERSION 2 – REVIEW

REVIEWER	Roger Ruiz Moral School of Medicine Universidad Francisco de Vitoria. Madrid. Spain
REVIEW RETURNED	08-Jan-2021

GENERAL COMMENTS	Manuscript ID bmjopen-2020-044753 entitled "The attitudes, perceptions and experiences of medical school applicants following the closure of schools and cancellation of public examinations in 2020 due to the COVID-19 pandemic".
---

	This is a revised version of the manuscript that follow most of the reviewers and editor suggestions. My main concern was about the presentation of the study, closer to a “dossier” or “report” than a standard paper. In this new version, authors have shortened the manuscript considerably (mainly cutting down the Introduction) and added more information into an Appendix. This has a positive impact upon the paper's 'readability'. I think the authors have made an important effort to better adapting to the BMJ Open recommendations for authors, without losing key information. Besides they included now several changes that increase the clarity and contribute to a better assessment of this study particularly into the Strengths and Limitations section of the Discussion. I think this manuscript can be accepted now for publication. Roger Ruiz Moral, MD, PhD
--	---

REVIEWER	Dr Don Munro School of Psychology, University of Newcastle, NSW 2308, Australia
REVIEW RETURNED	13-Jan-2021

GENERAL COMMENTS	As I said in my original review, this is a very valuable contribution to medical education and should be seen by deans and admission officers throughout UK. Your amendments to the paper have been well done, and it now reads much better and is more focussed. The only substantial change I'd like to see is a Conclusion that better shows the gist of the results (and perhaps some generalisations from them, such as an indication of what KINDS of things students felt were most unfair and most fair). At present it is too much taken up with platitudes about the general situation. I have also attached a copy of the amended paper, using Track Changes to comment on a few wording issues which you may care to address, but other than those points I am recommending publication at an early date. - The reviewer provided a marked copy with additional comments. Please contact the publisher for full details.
--

VERSION 2 – AUTHOR RESPONSE

Reviewer: 2

Please find additional comments from this reviewer in the attached file Dr. Roger Ruiz-Moral, School of Medicine. University Francisco de Vitoria. Madrid, Spain Comments to the Author:
Manuscript ID bmjopen-2020-044753 entitled "The attitudes, perceptions and experiences of medical school applicants following the closure of schools and cancellation of public examinations in 2020 due to the COVID-19 pandemic".

This is a revised version of the manuscript that follow most of the reviewers and editor suggestions.

My main concern was about the presentation of the study, closer to a “dossier” or “report” than a standard paper. In this new version, authors have shortened the manuscript considerably (mainly

cutting down the Introduction) and added more information into an Appendix. This has a positive impact upon the paper's 'readability'. I think the authors have made an important effort to better adapting to the BMJ Open recommendations for authors, without losing key information. Besides they included now several changes that increase the clarity and contribute to a better assessment of this study particularly into the Strengths and Limitations section of the Discussion.

I think this manuscript can be accepted now for publication.
Roger Ruiz Moral, MD, PhD

We thank the reviewer for his comments and are pleased he recommends publication.

We do not have additional comments in an attached file from Reviewer 2, but we do have them from Reviewer 3 (see below).

Reviewer: 3

Dr. D Munro, The University of Newcastle Comments to the Author:

As I said in my original review, this is a very valuable contribution to medical education and should be seen by deans and admission officers throughout UK. Your amendments to the paper have been well done, and it now reads much better and is more focussed. The only substantial change I'd like to see is a Conclusion that better shows the gist of the results (and perhaps some generalisations from them, such as an indication of what KINDS of things students felt were most unfair and most fair). At present it is too much taken up with platitudes about the general situation. I have also attached a copy of the amended paper, using Track Changes to comment on a few wording issues which you may care to address, but other than those points I am recommending publication at an early date.

Thank you, we are pleased the reviewer recommends publication. We have detailed below how we have responded to his comments in the attached file.

p4: change "states" to past tense.
Done.

p4: change "may" to past tense.
Done.

p7: change "performed in R" to "run on".
Done.

p16: change "and" to "though".
We have not changed this because it would imply 4 factors is better whereas it is just an alternative.

p19: change "likely" to "not unlikely/possible"
Done.

p20. Comma after (28).
Not done because we believe the convention is to have the comma before.

p20. In footnote 5, 'meet their offers' is unclear. Suggest 'are admitted' or similar. Meeting their offer is not the same as being admitted. As such, have changed to "Students meet their offer(s) and can be admitted if they achieve or exceed the grades specified."

p21: Needs improving – a large part is taken up by things that are common knowledge, but the main

conclusions from the evidence presented are absent.
Have included the phrase “, but which demonstrate the concerns many have about examination cancelations in 2020 and looking forward to 2021”.

We look forward to hearing from you shortly.

VERSION 3 – REVIEW

REVIEWER	Don Munro University of Newcastle, Australia
REVIEW RETURNED	20-Jan-2021
GENERAL COMMENTS	I am satisfied that all the recommendations in my previous review have been considered and/or acted on and that the paper is ready for publication.